

# Improving large-scale river routing models for climate studies: the impact of ESA long-term CCI discharge products on correcting multi-model hydrological simulations

Sadki Malak[1], Noual Gaëtan[2], Munier Simon[3], Pedinotti Vanessa[1], Verma Kaushlendra[3], Albergel Clément[4], Biancamaria Sylvain[5], and Andral Alice[2]

[1]Magellium, 1 Rue Ariane, 31520 Ramonville-Saint-Agne, France
[2]CLS, 11, rue Hermès, 31520 Ramonville Saint-Agne, France
[3]CNRM, Université de Toulouse, Météo-France, CNRS UMR 3589, Toulouse, France
[4]European Space Agency Climate Office, ECSAT, Harwell Campus, Didcot, Oxfordshire, United Kingdom
[5]Laboratoire d'Etudes en Géophysique et Océanographie Spatiales (LEGOS), Université de Toulouse, CNES/CNRS/IRD/UT3, Toulouse, France

**Correspondence:** Sadki Malak (malak.sadki@magellium.fr)

**Abstract.** Large scale hydrological models like CTRIP and MGB are essential for simulating river dynamics and supporting large-scale climate studies. Their accuracy can be significantly improved through satellite data assimilation. This study leverages 20 years of high-resolution discharge data (2000–2020) from the ESA Climate Change Initiative (CCI) to enhance CTRIP and MGB models via ensemble Kalman Filter frameworks (HyDAS and HYFAA). Applied to the Niger and Congo basins, the models assimilate discharge data derived from altimetry and multispectral imagery, alongside water surface elevation (WSE) anomaly data, to evaluate their impact on model performance.

Discharge assimilation was more effective than WSE anomaly assimilation, as it provided a more direct input for improving model accuracy. Temporal data density was the key factor in reducing bias and enhancing the simulation of seasonal flow patterns, with spatial coverage and data quality also playing important roles. In the Niger Basin, the assimilation of denser discharge data resulted in a significant bias reduction, which should improve the representation of long-term climate trends. Furthermore, the higher temporal resolution allowed for better capture of flow variability, which is crucial for both seasonal climate studies and short-term predictions, such as extreme hydrological events.

The study also emphasizes the trade-offs between data resolution and quality, particularly in the Congo Basin. Future advancements include merging altimetry and multispectral discharge data, improving the discharge retrieval algorithms using SWOT data, and refining data assimilation techniques to improve climate studies and river system modeling in complex, climate-impacted basins.

## 1 Introduction

Understanding river discharge and water storage is essential for accurately modeling the water cycle on a continental scale, especially in regions where in-situ hydrological data are sparse or unavailable. Large-scale hydrological models, such as ISBA-





CTRIP and MGB, are effective tools for simulating river hydrodynamics, but they are limited by uncertainties in input parameters and the simplifications needed to represent complex local processes at larger scales (Emery et al., 2017; Paiva et al., 2013). As these models evolve, integrating remotely sensed observations via data assimilation (DA) has emerged as a key method to reduce uncertainties and improve model performance (Wongchuig-Correa et al., 2020).

While a large number of studies have focused on the added value of water surface elevation (WSE) data assimilation into
models like ISBA-CTRIP and MGB (Pedinotti et al., 2014; Paiva et al., 2013; Oubanas et al., 2018), recent studies have started to explore the assimilation of discharge data into these models. For example, Paiva et al. (2013) applied discharge data alongside precipitation data from TRMM in the MGB model to improve simulations of river dynamics in the Amazon Basin. Similarly, Wongchuig-Correa et al. (2020) explored simulated SWOT discharge data for the Solimões and Negro river basins, emphasizing the potential of discharge assimilation in improving model accuracy. Additionally, Emery et al. (2018) used
altimetry-derived discharge data to enhance river storage and discharge simulations in large-scale models like ISBA-CTRIP, showing the growing interest in integrating discharge data for more accurate hydrological modeling.

The European Space Agency (ESA) recognized the need for longer-term and refined temporal data and initiated the ESA Climate Change Initiative (CCI) Discharge Project to address the lack of long-term, high-resolution river discharge data. Although many satellite-derived hydrological data products exist, they are often limited in temporal resolution and do not
cover long periods necessary for climate studies. The ESA CCI discharge products are designed to fill this gap by providing long-term, quasi-daily resolution data over two decades (2000–2020). This rich dataset offers the opportunity to capture the temporal dynamics of large river basins, allowing for the study of both seasonal and interannual variations, as well as extreme hydrological events. These high-resolution data over extended periods also enable the assessment of long-term trends and changes in river discharge patterns under varying climatic conditions, helping to understand the impacts of climate change
on hydrological systems by capturing variability and shifts in hydrological processes over time. Although this study does not address hydro-climatic aspects, it focuses on evaluating how the assimilation of these new long-term high-resolution data impacts model outputs and performance.

While existing research has explored the assimilation of long-term satellite-derived data, such as terrestrial water storage from missions like GRACE (Girotto et al., 2016; Getirana et al., 2017), these efforts have not specifically focused on assim-
ilating river discharge or water surface elevation (WSE) data, nor have they extended over two decades. The specific use of long-term WSE and discharge data in DA remains underexplored. Studies like Emery et al. (2018) have demonstrated the benefits of long-term data assimilation in reducing model bias and enhancing the accuracy of river discharge predictions. Previous research has highlighted that the success of DA depends not only on the accuracy of the satellite products but also on how well their spatial and temporal characteristics align with the model's scale and objectives (Oubanas et al., 2018; Emery et al., 2020).
For the Niger and Congo basins, where hydrological complexities abound, integrating these satellite-based discharge data into large-scale river routing models like ISBA-CTRIP and MGB presents both opportunities and challenges.

Building on these findings, the long-term assimilation of satellite-derived data, including WSE and discharge provided by the CCI products, is anticipated to lead to more stable and accurate hydrological simulations. This is achieved by effectively capturing both seasonal variations and extreme hydrological events over time. By addressing the gaps in existing





literature—specifically the lack of long-term (20 years) assimilation studies focused on discharge and WSE—this research leverages the superior temporal and spatial resolution of CCI products to significantly enhance model performance in large and complex river basins such as the Niger and Congo.

This study builds on the existing literature by focusing on the assimilation of the new long-term, high-resolution CCI discharge products into two large-scale river routing models, CTRIP and MGB. CTRIP is a continental-scale River Routing Model

(RRM), coupled to ISBA, a Land-Surface Model (LSM), that integrates large spatial scales with simplified processes, making it suitable for global-scale applications (Decharme et al., 2019a). In contrast, MGB is a distributed basin-scale hydrological model that provides more detailed spatial representation and is primarily used for flood forecasting and hydrological assessments at the basin level (Paiva et al., 2013). The key difference between these models lies in their scale and resolution: CTRIP operates at a coarser scale, focusing on large-scale hydrodynamics, whereas MGB focuses on local basin processes with a finer

spatial resolution.

In the context of data assimilation into large-scale hydrological models, satellite-based observations, particularly water surface elevation (WSE) and discharge, offer distinct advantages. WSE data, while having lower uncertainty, can present challenges when integrated into models like CTRIP and MGB due to mismatches with typical hydrological model outputs and simplified river geometries. In contrast, discharge data more closely aligns with model simulations, but the process of deriving

discharge from WSE, as one of the products of the CCI project (Gal et al., 2024), introduces additional uncertainties. The newly available discharge product derived from multispectral imagery (ESA CCI ATBD , 2023) carries even higher uncertainties due to the complexity of their production chain.

The primary objective of this study is to assess how these products, despite their varying levels of uncertainty, improve model performance when assimilated through DA. Specifically, we seek to address two key questions: (1) Does the assimilation of

higher uncertainty river discharge data improve model performance more than lower uncertainty water surface elevation data? (2) How do trade-offs between spatial coverage, temporal resolution, and the quality of satellite-derived products impact the effectiveness of DA for correcting hydrological model simulations across large basins?

The paper is organized as follows: Section 2 covers the study area and the models used, MGB and CTRIP, also providing a detailed explanation of the data assimilation approach and the satellite-derived CCI products that were assimilated. Section

3 explains the experimental setup. Section 4 presents the results, starting with the models' performance in Open-Loop mode and moving on to a duo-model analysis of the impact of assimilating CCI discharge products, including the added value of multispectral and altimetry discharge data. Section 5 explores the trade-offs between different satellite data types, focusing on spatial and temporal resolution and their influence on model accuracy. Finally, Section 6 concludes the paper, summarizing key findings and suggesting future research directions in the context of using long-term satellite data to improve large-scale

hydrological modeling.





## 2   Study domain, model and data used

### 2.1   Selected basins

Two basins were selected for this study: the Niger and the Congo basins. This choice relies on existing versions of the MGB model, which has been set up and calibrated in previous studies.

#### 2.1.1   Niger basin


Originating in the Guinean highlands, the Niger River is the third longest river in Africa (4,200 kilometres), crossing nine countries (Benin, Burkina Faso, Cameroon, Chad, Côte d'Ivoire, Guinea, Mali, Niger, and Nigeria) and providing water to about 100 million people (Figure 1). It ends in Nigeria, discharging through a massive delta into the Gulf of Guinea within the Atlantic Ocean. On its way through Mali, it traverses a vast floodplain called the Inner Delta, averaging 73,000 square kilometres, which dissipates a significant proportion of the water flowing in the river through absorption and evaporation (Andersen and Golitzen, 2005).


Downstream of the Delta, the river continues its course until Niamey, receiving water mainly from a set of three intermittent rivers in the Niger right bank which are responsible for a hardly predictable flood in Niamey, called the "red flood" (Cassé et al., 2015). On the left bank, most channels are not connected to the river system. The basin has an average active area of about 1.5 million square kilometers. The socio-economic situation of the Basin countries relies on many river-related activities (agricultural productions, freshwater fishing, and livestock farming), making water availability a critical issue for the development of its population.


The Niger basin traverses very different climatic zones, from the tropical humid Guinean coast, where it generally rains every month of the year, to the desertic Saharan region with very little precipitation. Its hydrological regime is highly dependent on the monsoon's rainfalls, whose uncertainties remain the main source of modeling errors in streamflow.





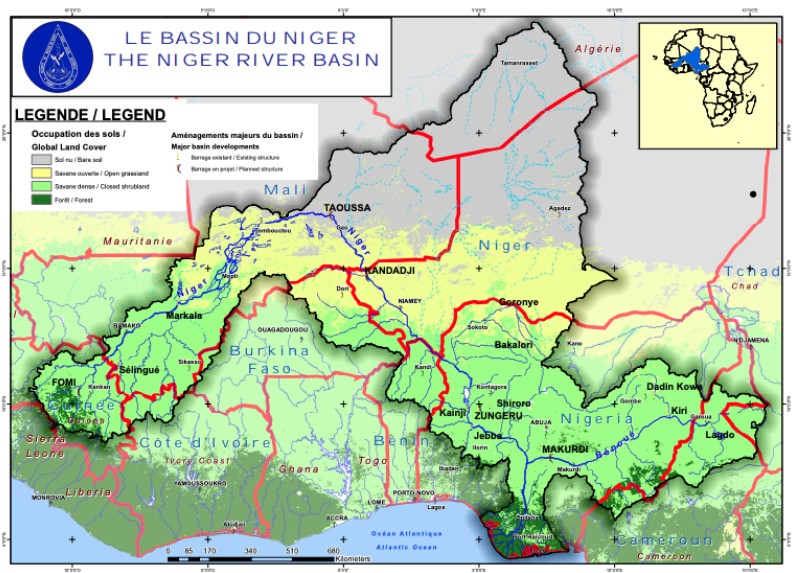

**Figure 1.** Map of the Niger River with the Niger River Basin in green (source : Niger Basin Authority)

### 2.1.2 Congo basin

The Congo River Basin (CRB) is the second-largest basin on earth and drains more than 3.7 million km$^2$. Despite this major contribution to the world's freshwater cycle, its hydrological behavior is not fully understood yet. The Congo's mean annual flow is around 41,000 m$^3$/s (Laraque et al., 2013). This mean flow is remarkably stable (Spencer et al., 2016), which makes it

an interesting singularity. Despite its critical importance to local, regional, and global water and carbon cycles, the CRB has not received as much attention as the Amazon or other large river basins.

In the CRB, most people rely on local resources, which are strongly impacted by climate change and water availability (Youssoufa Bele et al., 2013). Hence, the ongoing climate changes are expected to have severe implications for the populations. The mean temperature is approximately 25°C, whereas the mean annual rainfall varies between 2,000 mm/year in the central

parts of the basin and 1,000 mm/year both northward and southward. The annual potential evapotranspiration is around 1,000 mm/year and slightly varies across the basin.

The basin is characterized by four main drainage systems (Ubangi River in the northeast, Sangha River in the northwest, Kasai River in the southwest, and Lualaba River in the southeast) that converge to form the main Congo River (Figure 2).



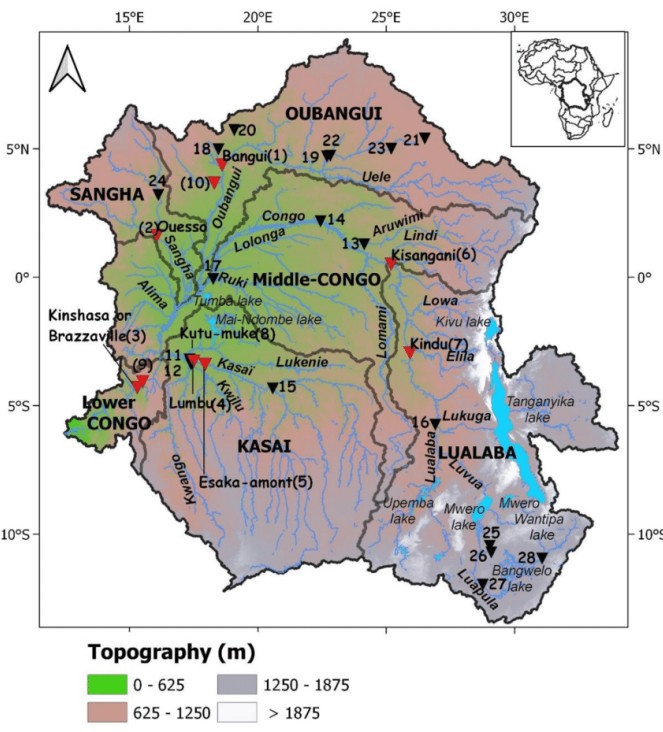

**Figure 2.** Topography of the Congo River basin derived from the MERIT digital elevation model. Red and black triangles represent, respectively, the gauge stations with current (> 1994) and historical observations (Kitambo et al., 2022).

## 2.2 Large-scale hydrological models

### 2.2.1 CTRIP global river routing model

The CTRIP model is a physically based river routing model coupled to the ISBA land surface model (Figure 3). While ISBA represents the vertical exchanges of water and energy at the soil-atmosphere interface, CTRIP simulates river flows over an entire hydrographic network (Decharme et al., 2019a). The ISBA and CTRIP models are used, in particular, in climate models such as the CNRM-CM6 that participated in the sixth phase of the Coupled Model Intercomparison Project (Eyring, 2016, CMIP6) as a contribution to the IPCC Sixth Assessment Report (AR6). CTRIP is based on a regular grid with a resolution of 1/12° (i.e. around 8 km at the Equator) obtained by the upscaling of the MERIT-Hydro global hydrographic network (Yamazaki et al., 2019), available at a resolution of 90 m and currently considered to be the most accurate on a global scale.

A number of hydro-geomorphological parameters, such as the lengths and slopes of river sections, are obtained from high-resolution data from MERIT-Hydro, while other parameters, such as widths, depths, and roughness, are obtained from empirical formulae (Munier and Decharme, 2022). It is assumed that each grid cell contains one and only one river reach, represented as a reservoir flowing into the downstream grid cell. The Manning equation is used to calculate the flow velocity as a function




of the volume of water in the section, itself updated by the inflows from the upstream reaches and the runoff from the ISBA model.

The CTRIP model also benefits from a two-dimensional representation of aquifer dynamics and groundwater-river exchanges (Vergnes and Decharme, 2012). Finally, surface processes linked to vegetation (including real evapotranspiration) and snow cover (including sublimation and melting) are taken into account in the ISBA model. It is also important to note that because the prior objective of CTRIP is to be integrated into climate models, it is based on quite simple approximations of rivers and aquifers dynamics. Consequently, unlike most hydrological models, the CTRIP model does not benefit from a parameter calibration stage. This choice ensures spatial consistency when the model is used in other regions of the world—or even globally—where few observations are available.

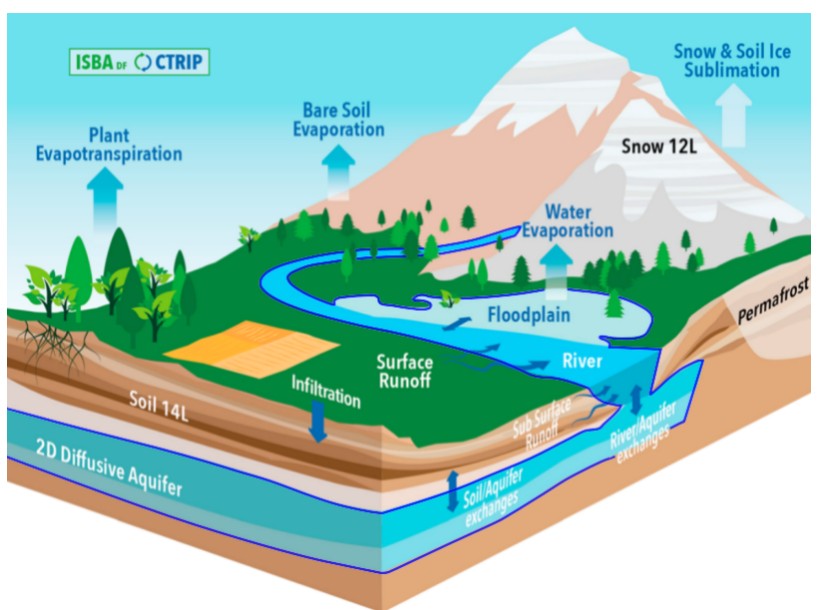

**Figure 3.** Schematic representation of the ISBA land surface model coupled to the CTRIP river routing model (Decharme et al., 2019a)

### 2.2.2 MGB regional hydrological model

The MGB model (Collischonn et al., 2007; Pontes et al., 2017) ("Modelo de Grandes Bacias" or "Large Basins Model") is a semi-distributed hydrological model that uses physically and conceptually based equations to simulate the continental phase of the hydrological cycle (Figure 4). It has been evaluated on large basins around the world and has shown good ability to represent complex processes, such as those governing the dynamics of flooded areas (Paiva et al., 2013). The particularity of this version is that it utilizes a two-way coupling scheme between hydrological and hydrodynamic modules, allowing for a more dynamic representation of wetlands.

Specifically, infiltration from flooded areas into unsaturated soil columns is accounted for, which is crucial in regions where dry soils are inundated by exogenous floodwaters, such as the Niger Inner Delta. Additionally, the model dynamically simulates

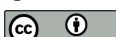



changes in evapotranspiration and open water evaporation based on land cover variations during flooding, capturing the differing effects of bare soil and vegetation across flooded zones. The routing function simulates the movement of water through the river network, accounting for flow propagation, backwater effects, and complex hydrodynamics in large floodplains. A full description of the model can be found in Fleischmann et al. (2018).

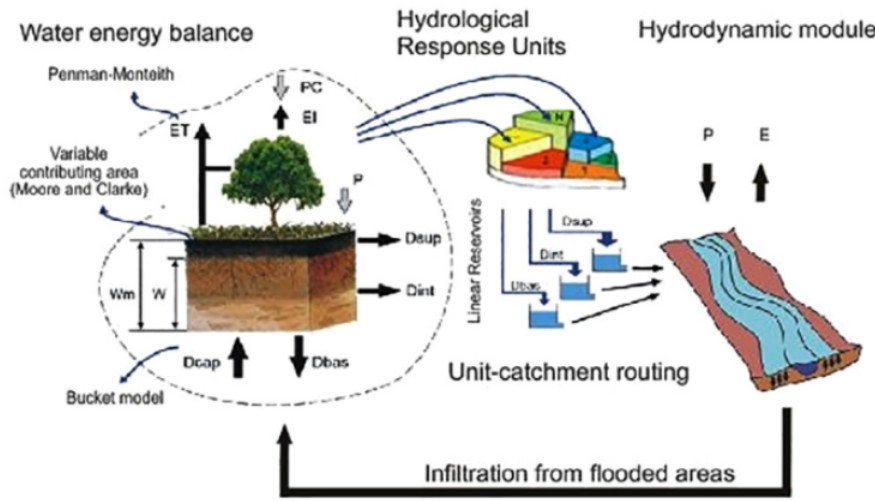

**Figure 4.** Main processes represented in MGB model ((Fleischmann et al., 2018)). PC: precipitation; ET: evapotranspiration; EI: evaporation of intercepted water; P : precipitation discounted by EI; Dsup: surface flow; Dint: sub-surface flow; Dbas: groundwater flow.

    MGB is used for real-time and near real-time flood forecasting in various regions, including the Amazon basin, where it has
been operational for flood forecasts and water resource management (Paiva et al., 2013; Siqueira et al., 2018; Fleischmann et al., 2020). Similar real-time forecasting applications have been implemented in the Paraná basin and other regions, showcasing the model's adaptability for hydrological forecasting at both local and large scales.

### 2.2.3   Models implementation over the Niger and Congo basins

    Both ISBA-CTRIP and MGB hydrological models require meteorological forcings, including precipitation, surface tempera-
ture and air humidity, wind, and radiation forcings.

    In this study, the ERA5 reanalysis was used to force the ISBA land surface model. The configuration of ISBA and CTRIP models is the one used for global scale simulations, as described in Munier and Decharme (2022), except that the floodplain scheme was not activated. It is important to note that no observations were used to calibrate the model parameters.

    The MGB model setup used in this study over the Niger basin is a resampled application of Fleischmann et al. (2018),
discretized into 11,595 unit-catchments and river reaches of approximately 10 km. The model was forced using the GSMAP precipitation dataset (Ushio et al., 2003), together with CRU 10′ long-term climatology data (monthly climate normals of





wind speed, solar radiation, relative humidity, air pressure, and air temperature) (New et al., 2002) for computation of model evapotranspiration. The model was calibrated against in-situ discharge time series and validated against in-situ discharges, satellite altimetry, and flooded areas.

In the Congo Basin, the model is discretized into 9,220 hydrological units of approximately 20 km$^2$ and is forced by IMERG rainfall data (Huffman et al., 2018).

    For both basins and both models, simulations are conducted over the period 2000–2020.

## 2.3   Data Assimilation schemes

### 2.3.1   CTRIP-HyDAS and MGB-HYFAA

CTRIP-HyDAS is a hydrological data assimilation system embedded into the CTRIP model. It has been developed since the early 2010s and is particularly designed to assimilate observations from satellite altimetry missions. The system has been applied over the Amazon basin (Emery et al., 2018, 2020a, b), either to assimilate nadir altimetry data or SWOT-like observations. With this system, it is possible to correct model states (mainly river storage) or parameters (such as roughness).

    In past studies, the assimilation of SWOT-like data was implemented in twin-experiment OSSEs since real SWOT obser-

vations were not available yet. Some pseudo-OSSEs have also been conducted by using a different model (namely MGB) to simulate SWOT observations (Figure 5), thus providing more realism in the differences between the dynamics of CTRIP and reality. CTRIP-HyDAS is currently being extended to be applied at the global scale at the 1/12° resolution in the context of the SWOT mission. Here, only states are corrected via the assimilation of either water surface elevation anomalies or river discharge. Assimilated data are those derived in this CCI project.

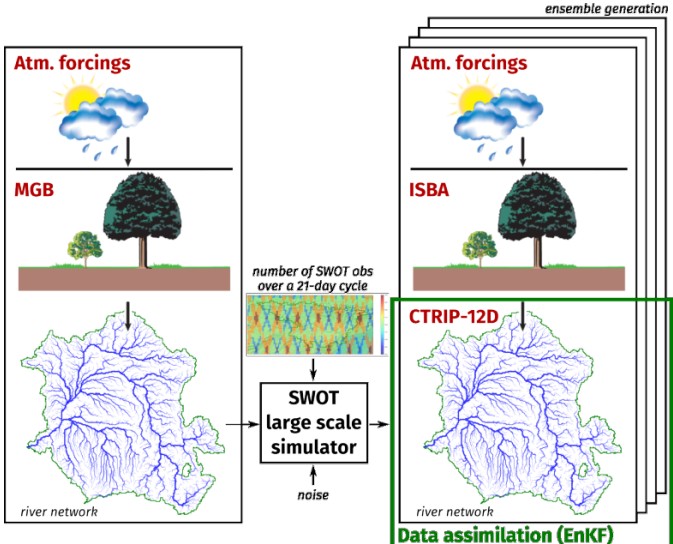

**Figure 5.** Schematic of the pseudo-OSSE to assess the CTRIP-HyDAS over the Congo basin. SWOT observations are simulated using the MGB model and a large scale simulator that accounts for realistic SWOT orbit and noise. Simulated SWOT data are then assimilated into





HYFAA (Figure 6) is a Python scheduler developed for hydrological monitoring and forecasting purposes. Its processing module manages the launch and communication between three components: the MGB hydrological model, an Ensemble Kalman Filter (EnKF), and an observations database. In its forecasting mode, the scheduler's tasks can be divided into three main steps:

1. Pre-processing: Updates local forcing and assimilation databases from external sources, converts data to the required
format, and keeps track of changes in data.

2. Processing : Launches hydrological simulations. When assimilation data is available, the processing module launches the EnKF to update the model state and/or parameters consistently with the observation and model uncertainties. The MGB model state variables and parameters are given as inputs to the EnKF, which returns corrected state variables and parameters as new inputs to the MGB model. The processing module manages these exchanges of data.

3. Post-processing : Keeps track of the simulated or analyzed model states in an SQL database.

The HYFAA platform has been used in various OSSE studies to assess the contribution of future satellite missions (SWOT, SMASH), as well as to evaluate the relevance of current observation systems, their spatial and temporal resolutions, and their uncertainties in estimating discharge over large river basins around the world.

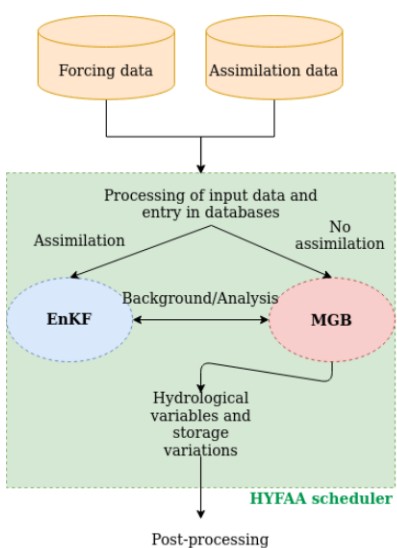

**Figure 6.** Conceptual scheme of the HYFAA platform architecture and components

### 2.3.2 The Ensemble Kalman Filter (EnKF)

The Kalman filter (Kalman, 1960) is a foundational sequential data assimilation technique extensively used in hydrological sciences. At each time step, it integrates observations with the forecasted state ensemble to produce an updated analysis state,





effectively addressing both observational and model errors. Originally developed for linear systems with Gaussian errors, the Kalman filter has been adapted for nonlinear models, with the Ensemble Kalman Filter (EnKF) being a notable extension (Evensen, 2009). In this study, we used LETKF, which is a variant of the EnKF enabling high reductions of numerical costs.

As in Revel et al. (2019), the analysis equation becomes:

$$X^a = X^f + E^f \left[ V D^{-1} V^T (HE^f)^T \left( \frac{R}{w} \right)^{-1} (Y^o - HX^f) + \sqrt{(m-1)} V D^{-1/2} V^T \right] \qquad (1)$$

where $X^a$ is the posterior state estimator (or assimilator), $X^f$ is the prior state estimator (or forecast), $Y^o$ is the observation (here, WSE or river discharge), $H$ is the observation operator, which is used to convert the model state to model equivalent of observations, $m$ is the size of the ensemble, $E^f$ is the prior state error covariance, which is obtained directly from the ensemble,

$R$ is the observation error covariance, determined from the uncertainty of the measurements, $w$ is the weighting term for the observation localization, and $VDV^T$ is given by:

$$VDV^T = (m-1)I + (HE^f)^T R^{-1} HE^f \qquad (2)$$

where $I$ is the unit matrix with dimension $m$. $VD^{-1}V^T$ and $VD^{-1/2}V^T$ can be calculated from the eigenvalue decomposition of $VDV^T$. It is important to highlight that $R$ is consistently assumed to be diagonal, indicating independence among

observation errors.

The localization, described by $w$ in the analysis equation, is used to limit the spatial influence of each observation. We used the same methodology as Revel et al. (2019) to compute the length of the localization for each pixel (or river reach) of the domain. The method is based on the semi-variogram from a long-term free run of the model. For each pixel of the domain, the basic idea is to compute the correlation between the discharge time series at this pixel and at pixels connected via the

river network (upstream and downstream). Then, with a given threshold, we delimitate the pixels that will be corrected if an observation is available at the original pixel. This localization method is more adapted to hydrology than more classical methods simply based on geographical distance or on the distance within the river network.

Figure 7 shows an example of the localization for some pixels in the Congo basin.





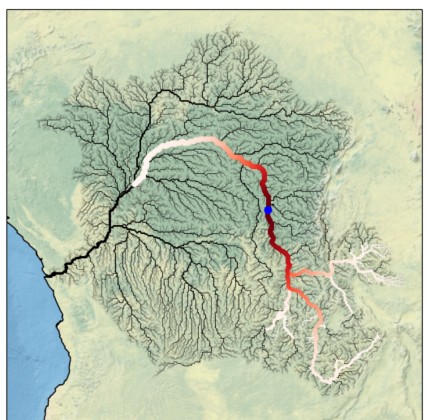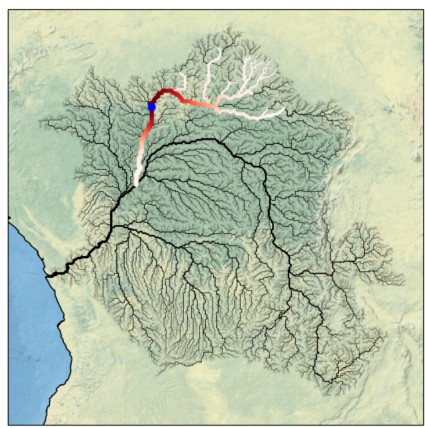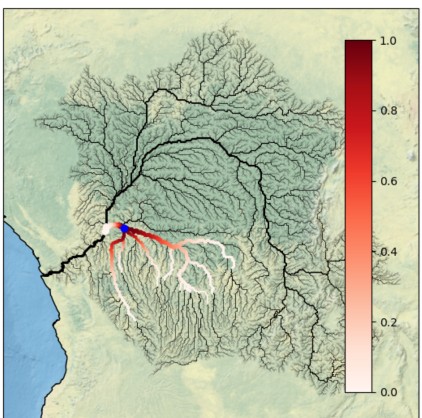

**Figure 7.** Examples of the localization for three pixels of the Congo basin

### 2.3.3 Ensemble Generation

The ensemble of model states, representing the best operating estimate, is systematically generated through the introduction of perturbed meteorological forcing. It is highlighted that only meteorological forcing is assumed to be subject to uncertainties, with the model structure and parameters assumed to be devoid of errors that could significantly impact the model outputs. Perturbing the model parameters is also a possible way to create the model ensemble, and it is particularly useful when parameters are corrected during the analysis step.

Here, we used the method developed in Munier et al. (2015). The approach involves perturbing statistically significant modes obtained through the decomposition of the precipitation field into Empirical Orthogonal Functions (EOFs). Mathematically, for a variable $X$ representing meteorological forcing, the decomposition can be expressed as:

$$X_{ijk} = X_{\text{ref},jk} + \sum_{l=1}^{\text{NEOFs}} U_{l,jk} V_{i,l} \tag{3}$$

Here, $X_{ijk}$ represents the meteorological forcing variable at time step $i$, latitude $j$, and longitude $k$. $X_{\text{ref},jk}$ is the temporal
mean of $X$ at latitude $j$ and longitude $k$. $U_{l,jk}$ denotes the spatial patterns, and $V_{i,l}$ represents the temporal coefficients obtained through the EOF analysis. The sum is taken over the retained EOFs, with NEOFs being the number of retained modes.

The leading modes, explaining 95% of the variance, are retained, with the perturbations applied as:

$$\tilde{X}_{ijk} = X_{\text{ref},jk} + \sum_{\ell=1}^{\text{NEOFs}} \left(U_{\ell,jk} + \sigma U_{\ell,jk}\right)\left(V_{i,\ell} + \sigma V_{i,\ell}\right) \tag{4}$$

Here, $X$ represents the perturbed variable, and $\sigma$ is the standard deviation of the respective EOF mode. This methodology
ensures a realistic representation of meteorological forcing uncertainties, critical for generating the ensemble of model states for subsequent assimilation experiments.





## 2.4 Observations

### 2.4.1 Assimilated satellite products

Within this study, three satellite-based products are evaluated within data assimilation schemes: altimetric water surface eleva-
tion (WSE), altimetry-based discharge (Qalti), and multispectral imagery-based discharge (Qmultispec) (Gal et al., 2024).

The first product, WSE time series, is derived from the merging of data from multiple nadir altimetry satellite missions
(Biancamaria et al., 2024), including ERS-1 and -2, Envisat, SARAL, Topex-Poseidon, Jason-1, -2 and -3, Sentinel-3A and
–3B, Sentinel-6A and Cryosat-2. Time series are derived at reference virtual stations (VS), preferably located on the Jason-3
ground track since the Jason series is the longest continuous series of altimeter missions. To ensure consistency across missions,
inter-mission biases are corrected by subtracting the mean difference between overlapping time series. In cases where missions
do not overlap, intermission bias is corrected by computing the difference between the time average of time series of the two
consecutive missions. Additionally, when VS from different missions are located within 10 km of each other, with no major
tributary between them, their time series can be merged using fitted relationships with the reference VS (?Biancamaria et al.,
2017).

The second product corresponds to river discharge derived from altimeter-based WSE (Qalti). Different methodologies have
been developed, but only the first one was used for data assimilated in this work. It relies on the existence of in-situ river
discharge observations and the calibration of a rating curve between in-situ discharge and altimetry WSE. A comprehensive
uncertainty analysis has been conducted to derive uncertainties associated with each discharge value. These uncertainties have
been used in the assimilation experiment, in comparison with constant values to assess the importance of providing realistic
uncertainties along with observations. For detailed methodologies and additional context regarding Qalti, refer to the Algorithm
Theoretical Basis Document (ATBD) by Gal et al. (2024) and the relevant published studies (Kouraev et al., 2004; Paris et al.,
2016; Zakharova et al., 2020, 2021; Tourian et al., 2013, 2017) which explore various approaches for deriving discharge from
satellite altimetry.

The third product corresponds to river discharge based on multispectral images (Qmultispec) Gal et al. (2024). The multi-
spectral images used comes from Landsat-5/7/8/9, TERRA and AQUA (MODIS sensors), Sentinel-2, and Sentinel-3 (OLCI)
platforms. It assumes that the differences between the passice response of the reflectance signal from the soil and that from the
water can be used to identify a change in the land area near the river channel that is strongly correlated with river discharge. In
this approach, different reflectance indices are computed for each considered VS where in-situ discharge is available. Then, the
estimation of discharge from these indices is very similar to the rating curve approach applied for the water levels by altimetry.
Here, the relationship is based on the evaluation of a non-linear regression relationship between the multi-mission time series
and the observed river discharge values. It is important to note that no uncertainties were computed in this approach, and only
constant theoretical error values were used in the assimilation experiment. Tarpanelli et al. (2013), Filippucci et al. (2022), and
Filippucci et al. (in preparation) provide more details on the generation of these products and the application of multispectral
imagery data for estimating river discharge.





Table 1 summarizes the available stations and data periods for the different CCI products used in the experiments for both
Niger and Congo basins.

**Table 1.** Data availability for WSE, Qalti, Qmultispectral products and in-situ discharge observations in Niger and Congo basins.

| Basin | Station | WSE (v1.1) / Qalti (v1.1) | Qmultispectral (v1.2) | Q in-situ |
|---|---|---|---|---|
| **Niger** | Mandiana | - | - | [2010–2018] |
| | Banankoro | - | - | [2010–2017] |
| | Koulikoro | [1995–2023] | [2000–2022] | [2010–2018] |
| | Ke-Macina | - | - | [2010–2018] |
| | Nantaka | - | - | [2010–2018] |
| | Akka | - | - | [2010–2017] |
| | Dire | - | - | [2010–2018] |
| | Ansongo | [1995–2023] | [2002–2022] | [2010–2018] |
| | Alcongui | - | - | [2010–2017] |
| | Niamey | [2017–2023] | [2000–2022] | [2010–2022] |
| | Malanville | [1995–2023] | [2013–2022] | [2010–2017] |
| | Makurdi (Benue) | - | - | [2010–2016] |
| | Ibi (Benue) | [2002–2023] | [2000–2022] | - |
| | Lokoja | [2010–2016] | [2000–2022] | [2010–2018] |
| **Congo** | Chembe-Ferry | [2002–2023] | [2002–2022] | - |
| | Ouesso | - | - | [1947–2020] |
| | Bangui / Oubangui | [2002–2023] | [2000–2022] | [1911–2020] |
| | Kinshasa | [2002–2023] | [2000–2022] | [1947–2023] |

The locations of the stations where data will be assimilated into both models are presented in the two maps shown in Figure8.

### 2.4.2 Comparison with in-situ observations

This section presents a comparison between the CCI discharge products and independent in-situ data from two different
sources. For the Niger Basin, we compare the CCI data with 13 stations from the Niger Basin Authority (ABN) network,
covering the period from approximately 2010 to 2017. In the Congo Basin, the comparison is made with the So-Hybam
database, which includes two stations that overlap with the CCI data, providing a longer temporal record from 1947 to 2020
(Table 1).

Figure 9 illustrates the performance of the KGE index and its three components — $\beta$ (bias), $r$ (correlation), and $\gamma$ (internal
variability) — for the CCI discharge products across stations in the Niger and Congo basins, with $Q_{\text{alti}}$ represented in blue and
$Q_{\text{multispec}}$ in green. Following this, Figure 10 presents the time series of discharge estimates from satellite altimetry ($Q_{\text{alti}}$) and
multispectral imagery ($Q_{\text{multispec}}$) compared to in-situ observations for five stations in the Niger basin and two stations in the
Congo basin, with an overview of the period from 2010 to 2020.





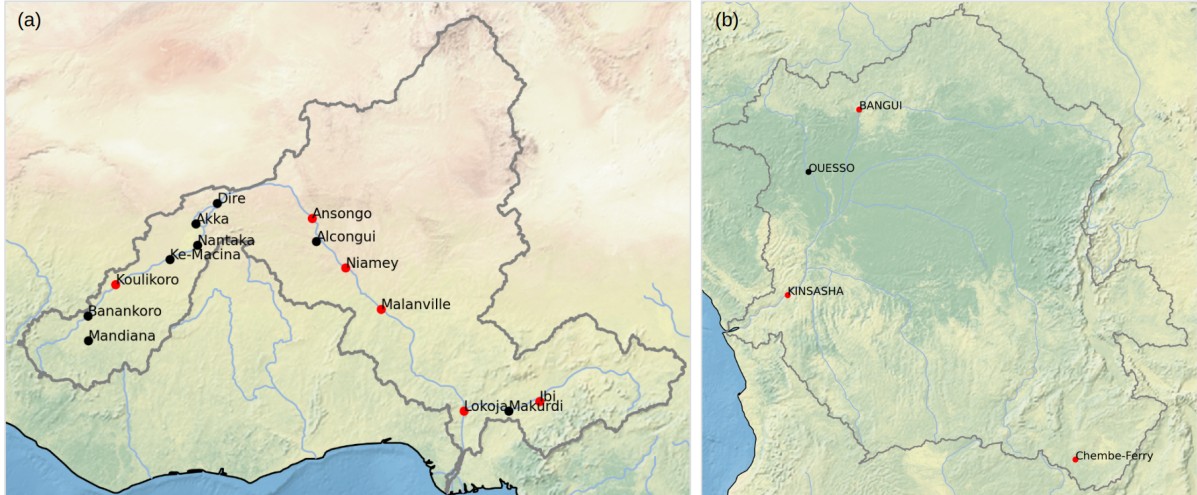

**Figure 8.** Maps showing the locations of stations in both the Niger and Congo basins. Red markers represent stations where data assimilation is conducted, and black markers indicate the remaining stations with in-situ data used for performance assessment of the assimilation experiments in both the MGB and CTRIP models.

In the Niger basin, the analysis of the 20-year time series (2000–2020) across stations such as Koulikoro, Ansongo, Niamey, Ibi, and Lokoja reveals distinct patterns in the performance of $Q_{\text{alti}}$ and $Q_{\text{multispec}}$ products. Figure 10 shows that $Q_{\text{alti}}$ data generally capture the seasonal hydrological cycle well, particularly in upstream stations like Koulikoro and Ansongo, where the correlations with in-situ data are 0.99 and 0.92, respectively. However, $Q_{\text{alti}}$ tends to overestimate high flows, as indicated by biases of 1.35 at Koulikoro and 1.06 at Ansongo. At Niamey, the availability of $Q_{\text{alti}}$ data starting only from 2017 limits the analysis. However, the two-peak hydrological cycle is reasonably well-represented, with a strong correlation of 0.96, although the peak flows are slightly underestimated (bias of 0.85).

Further downstream, Ibi, which is located on the Bénoué River (a significant tributary of the Niger River), shows a different pattern. Here, $Q_{\text{alti}}$ tends to underestimate discharge during both high and low flows, with a bias of 0.59. Despite this underestimation, the seasonality is well captured with a correlation of 0.99, as shown in Figure 10. At Lokoja, near the Niger River's outlet, a similar underestimation is observed with a bias of 0.90, but the seasonality remains well represented, with a correlation of 0.79.



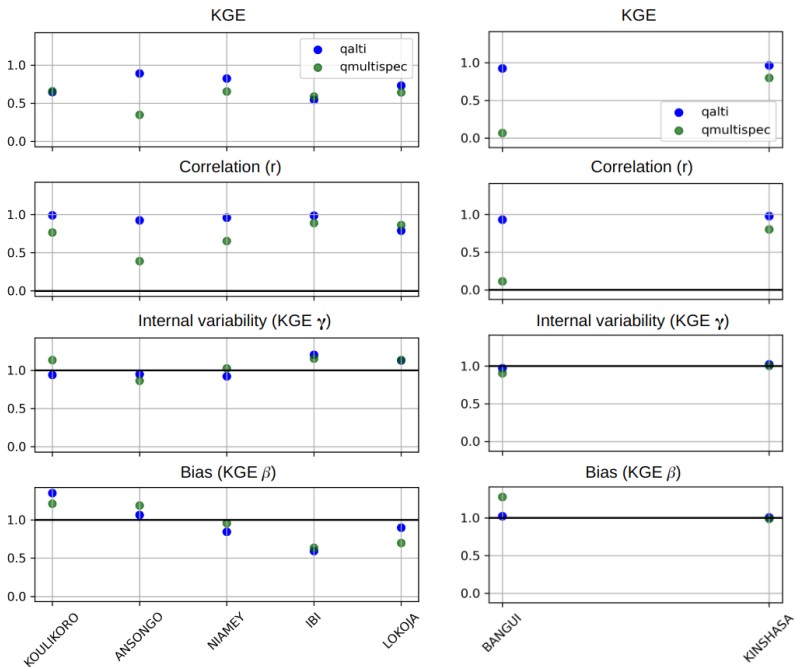

**Figure 9.** KGE score rate and its 3 components (r, $\beta$, $\gamma$) for CCI discharge products at stations where in-situ observations are available (Niger Basin left, Congo Basin right): altimetry-based discharge (Qalti) in blue and multispectral imagery discharge (Qmultispec) in green.

In contrast, $Q_{\mathrm{multispec}}$ data provide denser coverage, particularly during low-flow periods, as seen in Figure 10. However, this increased data density comes at the cost of increased noise, leading to generally lower performance scores compared to $Q_{\mathrm{alti}}$. For instance, at Ansongo, the correlation with in-situ data drops significantly to 0.40 for $Q_{\mathrm{multispec}}$, compared to 0.92 for $Q_{\mathrm{alti}}$. This decrease in performance is largely due to a temporal shift in the hydrological cycle representation, as well as greater data dispersion, particularly during high-flow periods.

At Koulikoro, $Q_{\mathrm{multispec}}$ data perform better in high-flow periods but exhibit a slight lag during the recession phase, resulting in a correlation of 0.76 compared to 0.92 for $Q_{\mathrm{alti}}$. Despite these issues, $Q_{\mathrm{multispec}}$ data show improved representation of low flows, closer to in-situ observations, compared to $Q_{\mathrm{alti}}$. Downstream at Lokoja, $Q_{\mathrm{multispec}}$ continues to underperform, with a bias of 0.70 compared to 0.90 for $Q_{\mathrm{alti}}$, and lower correlations due to increased noise and data dispersion.

At Niamey, the performance of the $Q_{\mathrm{multispec}}$ product against in-situ observations is subpar, with a KGE score of 0.65 for daily data, and dropping further to 0.57 for monthly data. This underperformance is primarily due to two factors: a temporal lag in the representation of the hydrological cycle and the failure to accurately reproduce the double peak characteristic of the flow at Niamey. The first peak, which results from the contributions of a nearby tributary, is particularly missed. This temporal lag significantly reduces the correlation, with a monthly correlation of only 0.62, compared to 0.97 for the $Q_{\mathrm{alti}}$ product derived from altimetry data.





In the Congo basin, where comparisons were limited to Bangui and Kinshasa due to the lack of independent in-situ data (not available at Chembe-Ferry), $Q_{\text{alti}}$ data show strong performance, closely aligning with in-situ observations. At both stations, $Q_{\text{alti}}$ achieved KGE scores exceeding 0.92, with correlations greater than 0.93 and minimal bias ( 1), demonstrating a robust representation of the internal variability of the hydrological cycle, as shown in Figure 9. In contrast, $Q_{\text{multispec}}$ data, while

showing similar correlation and internal variability representation at Kinshasa, suffer from greater data dispersion, leading to a lower daily correlation of 0.8 and a KGE score of 0.8.

At Bangui, however, the $Q_{\text{multispec}}$ data are less reliable, showing significant dispersion and failing to accurately represent the hydrological cycle. This is reflected in a very low correlation of 0.11 daily and 0.02 monthly, and a poor KGE score of 0.07, indicating that while $Q_{\text{multispec}}$ can offer denser temporal data, the quality and reliability are significantly compromised in

certain locations.

Despite the poor quality of the multispectral imagery discharge products, these data have been retained in the analysis for comparison purposes. Currently, the calibrated multispectral data is available only for Niamey in the Niger basin and for Kinshasa and Bangui in the Congo basin. This limitation constrains the analysis of the added value of data assimilation over the MGB model, considering the size of the basins, and restricts the ability to conduct a coherent comparison with other

experiments (dH, $Q_{\text{alti}}$). The calibrated versions are expected to be available in Phase 2 of the CCI-Discharge project, covering all analyzed stations. It will be valuable to revisit and reassess this work with the calibrated datasets once they become available to evaluate their performance.







**Figure 10.** Discharge time series comparing altimetry-based discharge (Qalti, blue), multispectral imagery discharge (Qmultispec, green), and in-situ observations (black) at five stations in the Niger Basin (first five panels) and two stations in the Congo Basin (last two panels).




## 3 Experimental Setup

To evaluate the impact of assimilating CCI discharge products Altimetry-based ($Q_{\mathrm{alti}}$) and Multispectral-imagery based ($Q_{\mathrm{multispec}}$)
and water surface elevation anomaly (dH) into the CTRIP (HyDAS scheme) and MGB (HyFAA scheme) models, a series of 8
experiments were conducted. First, both models were forced using ERA5 data (2000-2020), with MGB incorporating GSMAP
and IMERG precipitation datasets for the Niger and Congo basins, respectively (with unbiasing work done previously for
MGB), to establish a baseline in Open-Loop mode. A 25-member ensemble generated by the Ensemble Kalman Filter (EnKF)
scheme was used in both Open-Loop and assimilation experiments to account for uncertainties in input forcings.

Following the Open-Loop experiments, 8 assimilation experiments were carried out, with each CCI product tested at two-
three uncertainty levels:

- WSE anomaly (dH): computed errors, 0.2 m, 0.4 m.

- Altimetry-based discharge $Q_{\mathrm{alti}}$: computed errors, 30%, 50% constant uncertainty.

- Multispectral-imagery based discharge $Q_{\mathrm{multispec}}$: 30% and 50% constant uncertainty.

The performance of assimilation was evaluated against in-situ discharge data from the Niger Basin (ABN: 13 stations,
2010–2017) and the Congo Basin (So-HyBAM: 3 stations, 1950–2020).

The experimental setup is summarized in Figure 11, showing the models, forcing datasets, and the assimilation products
tested.

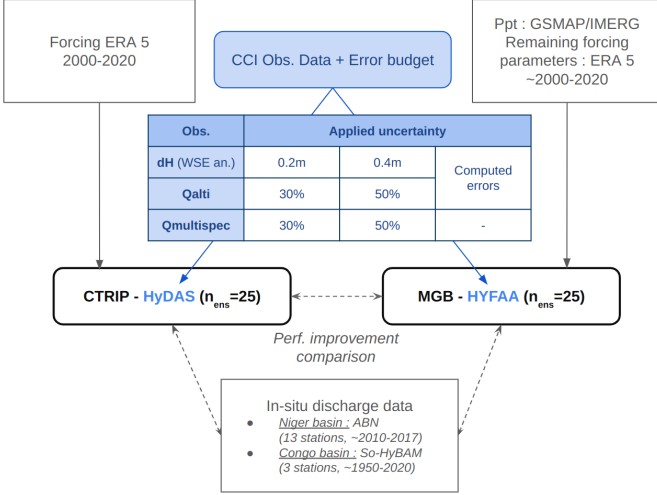

**Figure 11.** Schematic overview of the experimental setup for CCI discharge product assimilation into CTRIP and MGB models



# 4 Results

To correctly understand the impact of the assimilation of satellite derived WSE or discharge, it is first important to analyze the performances of both models without assimilation (Open-Loop), which is described in the following section.

## 4.1 Open-Loop performance

### 4.1.1 Niger Basin

The MGB model demonstrated overall good performance across the Niger Basin in its Open-Loop mode. It generally captured
the broad hydrological dynamics of the basin, with all stations—except Nantaka—showing a Nash-Sutcliffe Efficiency (NSE) score above 0.5. The correlation coefficient exceeded 0.8 basin-wide, with a median of 0.93, indicating the model successfully reproduced flow patterns during both high and low-flow periods. However, the MGB model tends to overestimate discharge during high-water periods, particularly upstream of stations like Koulikoro and Ansongo, with a median bias around 20%. The overestimation is particularly pronounced in the Inner Niger Delta, where it reaches 40% at Nantaka.

In contrast, the CTRIP model performs poorly across most of the basin, with a median NSE score of around -4 and values consistently below 0. This poor performance is primarily due to pronounced overestimations during high-flow periods. In the region between Ke-Macina and Niamey, discharge overestimations reach up to 2.3 times the observed median values, largely due to the model's inability to account for the significant evaporation and complex hydrodynamics of the Inner Niger Delta.

Moreover, CTRIP tends to underestimate discharge during low-flow periods, particularly along the Benue tributary and
near the Niger outlet at Lokoja. In contrast, the MGB model accurately captures low-flow conditions, especially in the downstream sections of the basin. However, MGB slightly underestimates internal variability, with a median KGE $\gamma$ value of 0.9, a discrepancy more evident in the central basin after the river passes through the Inner Delta.

In the Upper Niger (Banankoro, Mandiana, and Koulikoro), both models generally reproduce the basin's hydrological dynamics, capturing the seasonal peaks and low flows with correct timing. However, the key issue lies in the representation of
high-flow events. The MGB model performs strongly here, with NSE scores consistently above 0.65 and correlations exceeding 0.9. Meanwhile, the CTRIP model, although showing correlations above 0.8 in these stations, exhibits significant overestimation during high-flow periods, often doubling the observed high flows, as seen in panels (a) and (b) of the discharge time series (Figure 12).

In the Inner Niger Delta (Ke-Macina, Nantaka, Akka, Dire), both models encounter difficulties due to the region's complex
hydrodynamics, where the river splits into multiple branches and experiences significant water losses from evaporation. The MGB model overestimates discharge by over 20% during high flows, particularly at Nantaka, where the peak discharges are nearly doubled. The CTRIP model performs even worse, with exaggerated flow biases, particularly downstream of stations like Dire and Ansongo (panels (d,e) Figure 12). The overestimation grows from twice the observed values at Ke-Macina to three times at Ansongo. The failure to represent evaporation and the delta's complex branch dynamics leads to these inflated dis-
charge estimates in CTRIP. Additionally, CTRIP struggles to capture the gradual shift in the high-water season from September



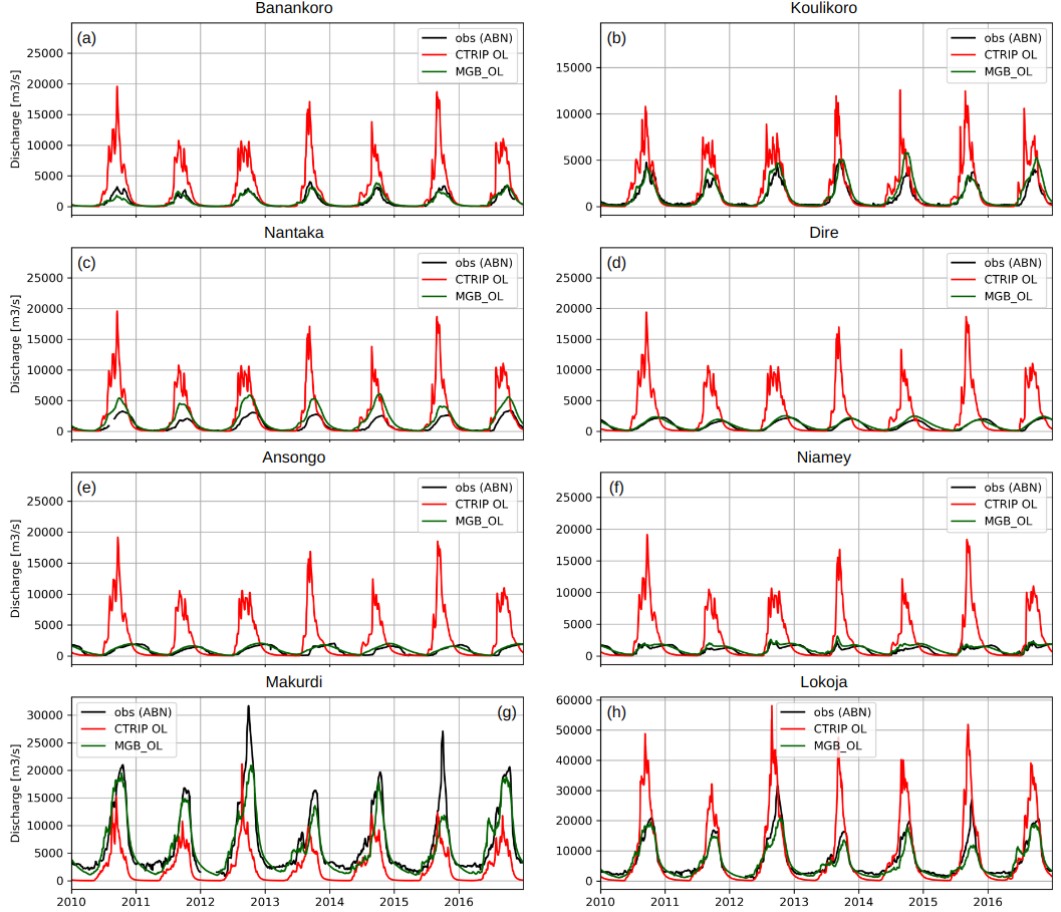

**Figure 12.** Discharge time series at same 8 stations within the Niger basin, comparing CTRIP (red) to MGB Open-Loop (green) and in-situ observations

to December/January, particularly between Nantaka (panel c) and Niamey (panel f), where the correlation drops below 0.5 as the flow moves downstream.

In the Middle Niger (Ansongo, Alcongui, Niamey, Malanville), the MGB model continues to show strong performance, particularly at Niamey, where it captures the bimodal hydrological cycle driven by the Sirba tributary (panel (c), Figure 13) (NSE = 0.8, r = 0.96). However, MGB overestimates the second flow peak by 20%. The CTRIP model continues to struggle in this region, with difficulties in capturing both the magnitude and timing of high flows. In Alcongui (Gorouol tributary), both models encounter challenges due to intermittent flows driven by seasonal rainfall, with correlation scores of r close to 0.8 in MGB and r close to 0.5 in CTRIP.

In the Lower Niger (Makurdi on the Benue tributary, and Lokoja near the Niger outlet), CTRIP performs relatively well compared to its performance upstream. While it underestimates discharge during both high and low flows, the hydrological



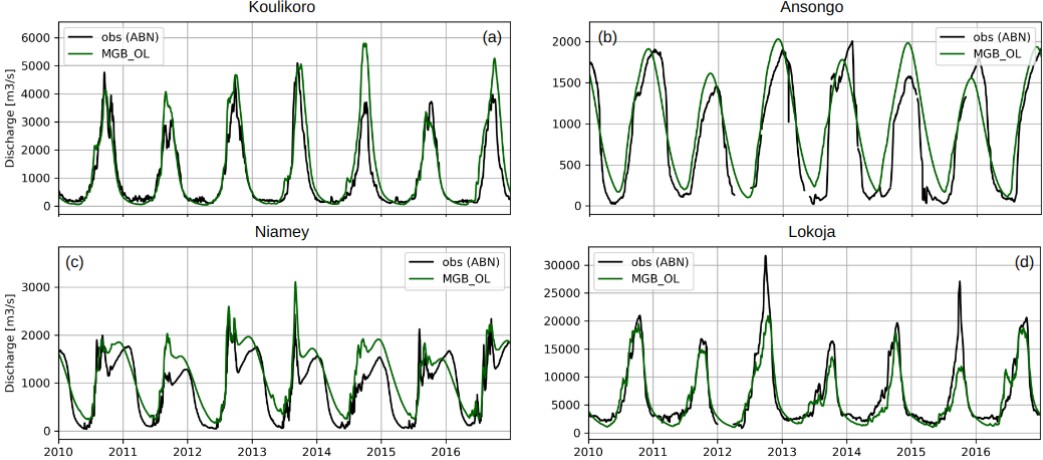

**Figure 13.** Discharge time series comparing MGB (green) to in-situ observations on Koulikoro, Ansongo, Niamey and Lokoja (Niger basin)

cycle is reasonably well reproduced at Lokoja (panel h, Figure 12). This is largely because CTRIP's parameters—such as channel width, slope, and Manning's roughness coefficient—are globally adjusted based on empirical relationships and global datasets to ensure consistent performance at the outlets of large river basins without region-specific calibration (Decharme et al., 2019b). However, CTRIP still struggles with high flow overestimation and low flow underestimation. The MGB model

performs better at Lokoja (panel d, Figure 13), as it is calibrated for each specific basin to accurately capture local hydrological processes. While the hydrological cycle is accurately reproduced, MGB tends to underestimate discharge during high flows, with about 10% underestimation at Lokoja, likely due to difficulties in fully representing the Benue's high flows.

### 4.1.2 Congo Basin

Considering the significantly limited access to in-situ station data across the Congo Basin, the performance analysis of the

MGB and CTRIP models remains constrained. The available in-situ data from the So-HyBam database includes only three stations, which cover two of the four main drainage systems: Ouesso on the Sangha tributary and Bangui on the Ubangui tributary. The third station is located at Kinshasa, downstream of the confluence of the four tributaries, on the Congo river. The models are therefore evaluated based on their performance at these three stations.

    The MGB model shows overall good performance in reproducing the hydrological dynamics of the Congo Basin, though

certain biases and errors persist. With a median correlation score of 0.88, the model successfully captures the general flow patterns across the three stations. The best performance is observed at Bangui, with a correlation of 0.94, while Ouesso and Kinshasa also perform consistently, with correlation values exceeding 0.87. However, the Nash-Sutcliffe Efficiency (NSE), with a median score of 0.4, suggests moderate accuracy in predicting discharge magnitudes, pointing to areas where the model could improve, especially in terms of magnitude prediction.





On a monthly scale, the MGB model generally performs well, particularly in terms of correlation (r), internal variability ($\gamma$), and NSE. This indicates good internal consistency and a strong ability to capture the seasonal and intra-annual variability in discharge. At Bangui and Kinshasa, KGE $\gamma$ monthly values are close to 1, reflecting the model's strength in reproducing seasonal cycles (panels d,f, Figure 14). However, short-term hydrological fluctuations present more challenges, as evidenced by the increased variability in daily scores, particularly at Ouesso (panel a, Figure 14). Here, the model overestimates internal

variability by around 20%, especially in the first peak of the bimodal cycle, highlighting difficulties in predicting short-term fluctuations.

     In contrast, the CTRIP model exhibits weaker overall performance, particularly near the outlet at Kinshasa, where NSE score falls below -4. On the tributaries, the model shows slightly better performances, with NSE values ranging between -0.7 and 0.3. The median correlation coefficient of 0.8 suggests that while CTRIP accurately captures seasonal flow patterns, the model

struggles with overall discharge magnitudes. Discharge is largely underestimated, with a median bias of 43% across the basin, particularly near Kinshasa (panels e,f, Figure 14).

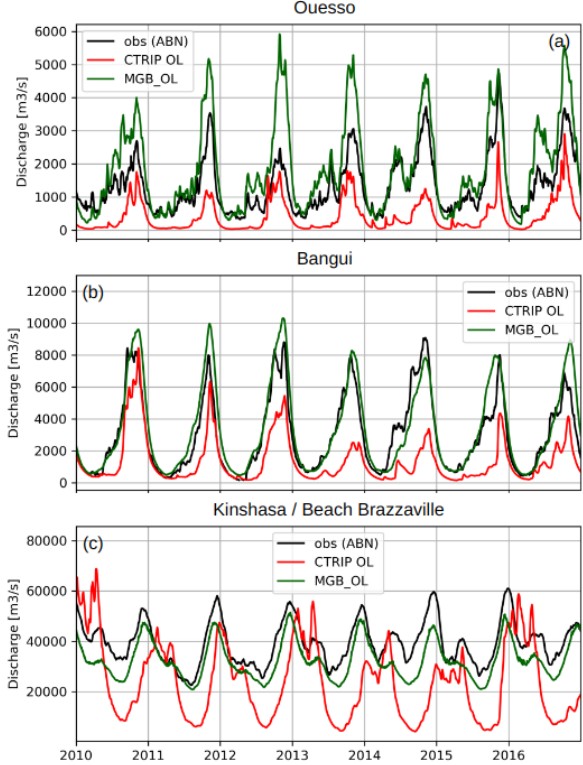

**Figure 14.** Discharge time series over the period 2010-2017 and climatologies comparing CTRIP (red), MGB (green), and in-situ observations (black) at three stations: Ouesso (a, b), Bangui (c, d), and Beach Brazzaville/Kinshasa (e, f)





At the tributary stations (Ouesso and Bangui), both models demonstrate reasonable performance in capturing the evolution of the annual hydrological cycle. The MGB model tends to overestimate discharge during high-flow periods, with overestimations of 17% at Ouesso (Sangha tributary) and 8% at Bangui (Ubangi tributary). For CTRIP, the annual cycle is well represented at both tributaries, but the model consistently underestimates discharge. The relative bias is 0.56 at Bangui and 0.32 at Ouesso, indicating a clear underestimation (panels b, d). Despite this, the correlation coefficients are satisfactory, with values close to 0.8 at both stations, reflecting a fairly accurate representation of the hydrological cycle.

At Kinshasa, the MGB model underestimates discharge by 19% across both high and low-flow periods. This underestimation is most pronounced during the second peak of the bimodal cycle (April to June), which is significantly underrepresented in the model, leading to an NSE score of 0 at Kinshasa. For CTRIP, the underestimation of discharge at Kinshasa is less pronounced compared to the tributaries, especially between February and April, when discharge is reasonably well captured. However, the model consistently underestimates discharge in the following months, with a relative bias of 0.63, likely due to underestimations of rainfall within the catchment area, leading to a shortfall in the observed water input and contributing to the systematic bias in discharge predictions at this station (Figure 14, panels e, f). Additionally, the correlation coefficient at Kinshasa drops to 0.46.

### 4.1.3 Key issues in both models

The Open-Loop performance of the MGB and CTRIP models across the Niger and Congo basins reveals several key limitations. MGB generally performs well but tends to overestimate discharge during high-flow periods in both basins, particularly in the Inner Niger Delta and at Ouesso and Bangui in the Congo basin. It also slightly underestimates internal variability, leading to difficulties in reproducing short-term fluctuations, especially in the central Niger Basin. Additionally, at Kinshasa, the model underestimates discharge by 19%, indicating a bias in the overall flow magnitude.

In contrast, CTRIP faces significant challenges, with large overestimations during high flows—up to 2.3 times the observed values in the Niger Basin—and considerable underestimations, particularly during low flows, across both basins. In the Congo basin, CTRIP struggles with discharge underestimations of around 43% at Kinshasa, although it captures seasonal flow patterns relatively well in the tributaries. It has to be noted that improvements can be expected with the CTRIP model when coupled to the ISBA land surface model to account for evaporation, namely. Yet, as CTRIP is also used for large scale hydrological studies, we preferred to use this configuration in the present study.

These model issues highlight opportunities for improvement, where data assimilation has the potential to reduce biases, improve discharge magnitudes, and refine flow timing in both models. The next section explores how different assimilation products can address these challenges and enhance model performance, taking into account the quality and temporal resolution of the assimilated products (Section 2.4.2), and also the uncertainties set to the different data (Section 3).





## 4.2 Assimilation Impact

### 4.2.1 Overview

The impact of data assimilation for both models (MGB-HYFAA and CTRIP-HyDAS) across the Niger and Congo basins is
first presented through a general overview of the performance changes observed across the different assimilation experiments.

To ensure the clarity of the figures, only the ensemble means for each experiment are considered, given the number of assimilation experiments. It is worth noting that, for both CTRIP and MGB, the dispersion between ensemble members (standard deviation) has consistently decreased following data assimilation compared to the Open-Loop simulations.

The following initial comparison highlights the overall trends and improvements brought by the integration of observation products, which will then lead into a more detailed analysis in the following subsections.

In the Niger basin, when assessing the performance of MGB in assimilating discharge and WSE anomalies, Qalti shows the highest improvement in NSE (Figure 15, panel 1a), with a median value of 0.83 at 50% uncertainty, followed by WSE anomalies (NSE: 0.73-0.76). In terms of correlation, Qalti assimilation performs best, raising the median correlation to 0.94 (Panel 1b), while WSE anomalies maintain a similar value to the Open Loop (0.93). Although Qmultispec exhibits the lowest NSE (0.68-0.72) and correlation due to its higher data noisiness, it compensates by performing best in reducing bias and correcting internal variability (panels 1c,1d). Bias is best reduced by Qmultispec, improving the median from 1.2 to closer to 1 (Figure 15, panel 1c). In terms of internal variability, Qmultispec also outperforms, with KGE $\gamma$ nearing 1.0 at 30-50% uncertainty, while Qalti and WSE anomalies range between 0.91-0.94 (Figure 15, panel 1d). Therefore, while Qalti leads in improving flow magnitude and correlation, Qmultispec proves superior in bias and internal variability correction.







**Figure 15.** Distribution of NSE (a), correlation (b), bias (c), and internal variability (KGE $\gamma$, panel d) scores for MGB (1) and CTRIP (2) simulated discharge at 11 in-situ stations (Niger Basin, 2000-2020): comparison of open-loop (grey) and assimilation of water level anomalies (dH, red), altimetry-based discharge (Qalti, blue), and multispectral imagery discharge (Qmultispec, green) with different uncertainties. The lightest colours for dH and Qalti experiments represent uncertainties from computed errors, followed by the lowest values, while the darkest indicate the highest errors. For Qmultispec, the shades of green transition from light to dark based on increasing theoretical uncertainties.





In CTRIP-HyDAS, the Niger basin exhibits limited improvement due to the model's already poor performance in Open Loop. However, Qmultispec performs better than both Qalti and WSE anomalies in bias reduction and overall improvement of internal variability. The CTRIP model demonstrates improvements in NSE, where Qmultispec assimilation leads to a 15% increase in performance, whereas WSE and Qalti show more modest gains in both bias and correlation (panels 2a, 2b, 2c, 2d, Figure 15).

For the Congo basin, assimilation experiments reveal a more nuanced picture. The analysis is limited to only two of the four main tributaries and Kinshasa at the lower Congo, making the scope of conclusions more constrained. While MGB displays noticeable improvements with Qmultispec in the Niger basin, the product does not perform as well in the Congo basin, largely due to lower data quality and temporal density (Figure 10). Here, Qalti delivers the best performance, notably improving NSE from 0 to 0.1 at Kinshasa. Correlation scores improve by about 5% at Kinshasa and other stations (Panels 1a, 1b, Figure 16).

However, bias and internal variability are only slightly corrected (less than 5% overall improvement), as seen in panels 1c and 1d of Figure 16.





**Figure 16.** Distribution of NSE (a), correlation (b), bias (c), and internal variability (KGE $\gamma$, panel d) scores for MGB (1) and CTRIP (2) simulated discharge at the 3 in-situ stations (Congo Basin, 2000-2020): comparison of open-loop (grey) and assimilation of water level anomalies (dH, red), altimetry-based discharge (Qalti, blue), and multispectral imagery discharge (Qmultispec, green) with different uncertainties.





Meanwhile, in CTRIP, the assimilation of WSE anomalies leads to better performance over the Congo basin compared to discharge assimilation. This is due to a more coherent alignment between the model's rating curve and observed WSE data (as shown in the following section 4.2.2), along with the lower uncertainty attributed to WSE compared to discharge. Panels 2b and 2d of Figure 16 show improvements in correlation (+13%) and variability (-7%), particularly noted at Kinshasa, where Qmultispec underperforms. In contrast, Qmultispec data quality deteriorates performance in both MGB and CTRIP for the Congo basin, particularly at Bangui and Kinshasa, where it fails to accurately represent the hydrological cycle.

When comparing the influence of the observations characteristics (spatial density, temporal frequency, and data accuracy) with uncertainty values, the distribution of performance indices across experiments shows that the characteristics of observed data play a more decisive role in both models and basins. Data assimilation is primarily driven by the spatial density of virtual stations, followed by temporal resolution and data accuracy, with uncertainty having a lesser impact. However, uncertainties become important in the presence of outliers, as higher uncertainty values reduce the likelihood of outlier assimilation, minimizing hashed effects in the analysis. This is evident in the assimilation of Qmultispec products in the Congo basin, where poor data quality degrades model performance. Nevertheless, higher uncertainty (50%-60%) limits degradation compared to 30% uncertainty (Figure 16).

From these results, two key findings are highlighted to explore more specifically the contributions of the different CCI discharge products. In one hand, the comparison between discharge assimilation and WSE anomaly assimilation in large-scale hydrological models is examined. In the other hand, the added value of Qmultispec data is analyzed, particularly in the Niger Basin, where it outperforms Qalti in correcting bias and internal variability.

### 4.2.2 Assimilation of Discharge Compared to WSE Anomalies in Large-Scale Hydrological Models

In the Niger Basin, MGB shows a clear advantage for discharge assimilation (Qalti) over water surface elevation anomalies (dH). Qalti improves Nash-Sutcliffe Efficiency (NSE) to 0.83 and raises correlation to 0.94, ensuring smoother flow predictions across stations.

As shown in Figure 17, at Ansongo and Malanville, discharge assimilation produces smoother results, while WSE introduces noise due to rating curve discrepancies between MGB and CCI observations. The hashed effect seen with dH assimilation is evident in several stations but absent in others like Ibi and Lokoja. This absence is due to the combined effect of low observation uncertainties and compatible rating curves between the model and observations at these stations.

This discrepancy is particularly evident at Ansongo (Figure 17), where MGB's rating curve diverges significantly from the CCI observations. Figure 18 highlights this finding by comparing observed and simulated rating curves across both Niger Basin stations. This mismatch between the two curves at Ansongo implies additional errors when converting WSE anomalies to discharge, leading to noisy results. In contrast, Qalti, as a directly modelled variable, fits more naturally to the hydrological model, avoiding these complexities and providing more reliable results in terms of discharge simulation.

At Ansongo, the MGB model results for two experiments (dH 0.4m and Qalti 30%) from 2010-2020 reveal additionnal differences in assimilation, as shown in Figure 19. Panels display time series for open-loop (black), Analysis (blue), and Observation to be assimilated (red). With dH assimilation, the MGB model shows near-zero residuals (otherwise the difference



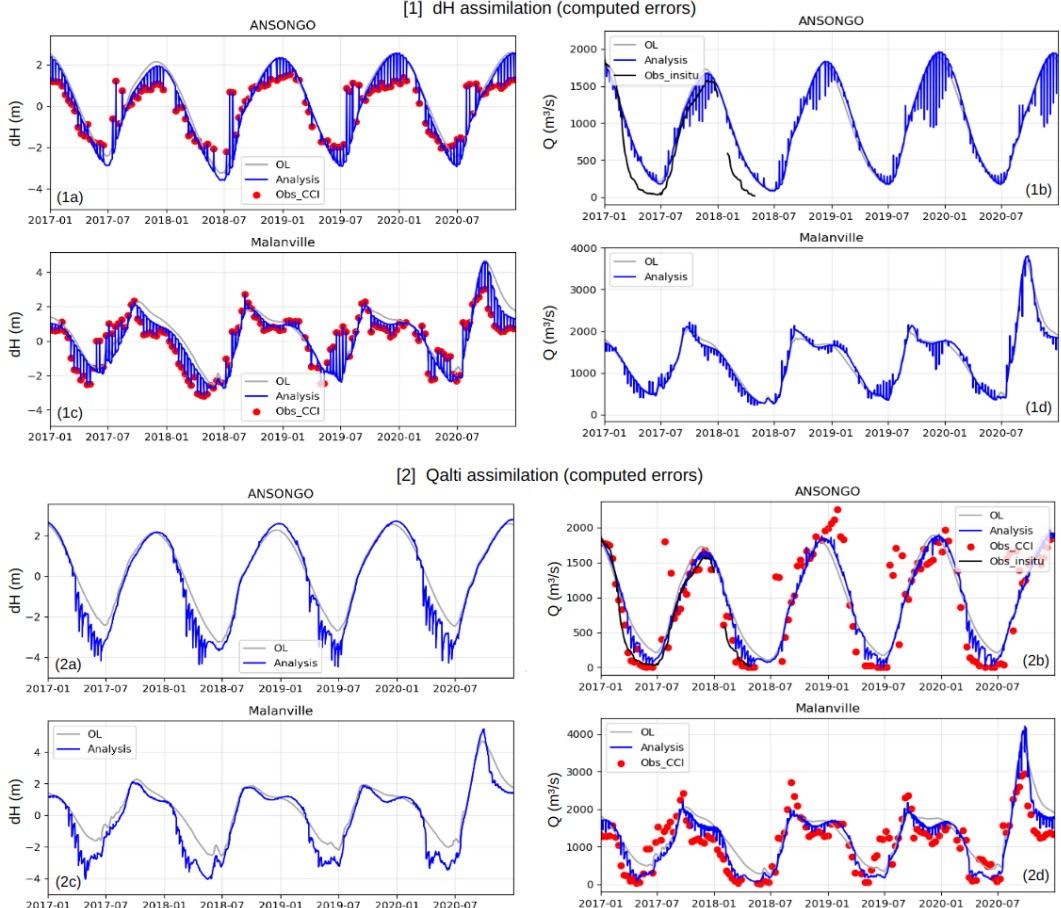

**Figure 17.** Comparison of water surface elevation (dH) and altimetry-based discharge (Qalti) assimilation with computed errors for Ansongo and Malanville stations (2017-2020), within MGB-HYFAA. Panel (1a-d): dH assimilation (computed errors) showing time series of dH (1a, 1c) and discharge (Q) (1b, 1d) for Ansongo and Malanville. Panel (2a-d): Qalti assimilation (computed errors) showing time series of dH (2a, 2c) and discharge (Q) (2b, 2d) for the same stations.

between the CCI observations assimilated and the Analysis), meaning all observations are assimilated with minimal corrections. The same pattern is observed for CTRIP. Due to their lower uncertainty compared to the other products, dH observations, even at 0.4m uncertainty value, are still assimilated even when they diverge from the model's theoretical WSE values. This introduces erroneous information into the model, which can disrupt the simulated discharge and amplify the hashed effect after

520 assimilation.

Qalti, on the other hand, even with 30% uncertainty, leaves persistent residuals, especially during high flow periods, and this is mainly because the error given to the discharge products is proportional, so the higher the discharge value, the higher the absolute uncertainty given to that value and the lower is the weight given to observation within the EnKF scheme compared to





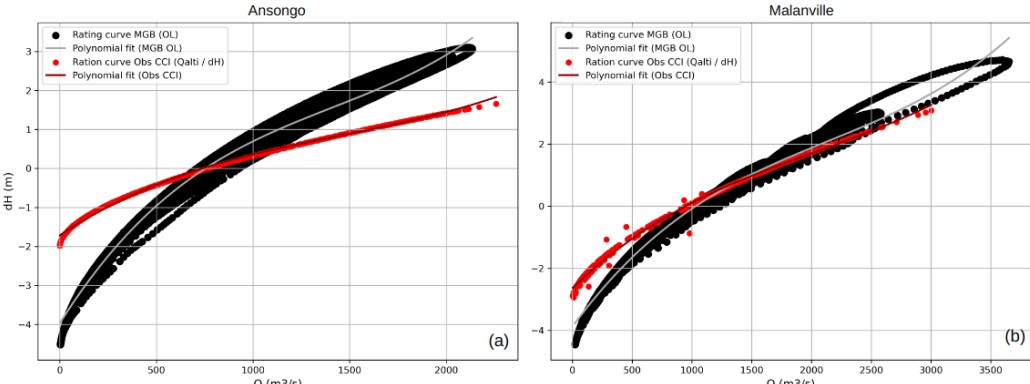

**Figure 18.** Rating curves from MGB (Open-Loop) vs. Obs CCI (dH, Qalti) at Ansongo and Malanville stations in the Niger basin.

model background. These residuals help maintain more reliable results by preventing the full assimilation of outliers (Figure 19).

In the Congo Basin, the discrepancy between the MGB model's rating curve and the observed altimetry data is even more pronounced, especially at Kinshasa. Here, Qalti further outperforms dH, as dH assimilation struggles with large water level variations not well captured by the model's flat rating curve (Figure 20). At Kinshasa, the MGB model has a flatter slope, meaning that for similar discharges, the model produces much smaller changes in water surface elevation. In contrast, the observed rating curve from altimetry (dH and Qalti) shows a much steeper slope, indicating a larger variation in water surface elevation for the same discharge fluctuations. When altimetry observations show dH variations reaching as high as ±7 meters, the MGB model's dH remains near zero.

This mismatch causes EnKF divergence and so outliers and very negative NSE values, making dH assimilation unreliable in stations like Kinshasa and Ouesso (Figure 21). Ouesso, correlated with Kinshasa through the localization matrix, is also impacted by the discrepancies observed at Kinshasa. These inconsistencies propagate backward to Ouesso, resulting in unstable discharge estimates with extreme fluctuations, including both excessively high and negative values. These issues lead to poor performance scores at Ouesso as well. It is worth noting that, in this version of MGB-HYFAA, the EnKF does not impose numerical constraints on discharge, which allows for negative values when unrealistic corrections deviate from physical plausibility, as seen in Figure 21.

In CTRIP, a contrasting result is observed. The assimilation of dH consistently outperforms Qalti across the stations in both basins. This difference is most notable in the Niger Basin, where the Open-Loop (OL) simulation performs poorly. Given the low performance of CTRIP's OL, any satellite product leads to significant improvements. The uncertainty for WSE anomalies (dH) corresponds to much lower uncertainty levels compared to Qalti, which is logical since the discharge is a more indirect product. Also, in CTRIP, for most stations, the model rating curve matches the observed one from altimetry products (dH/Qalti). However, for products with equal temporal density, the lower uncertainty assigned to dH (0.2m-0.4m) compared to Qalti (30%-50%) allows dH to provide more impactful corrections to the model. CTRIP model's rating curves along stations align closely





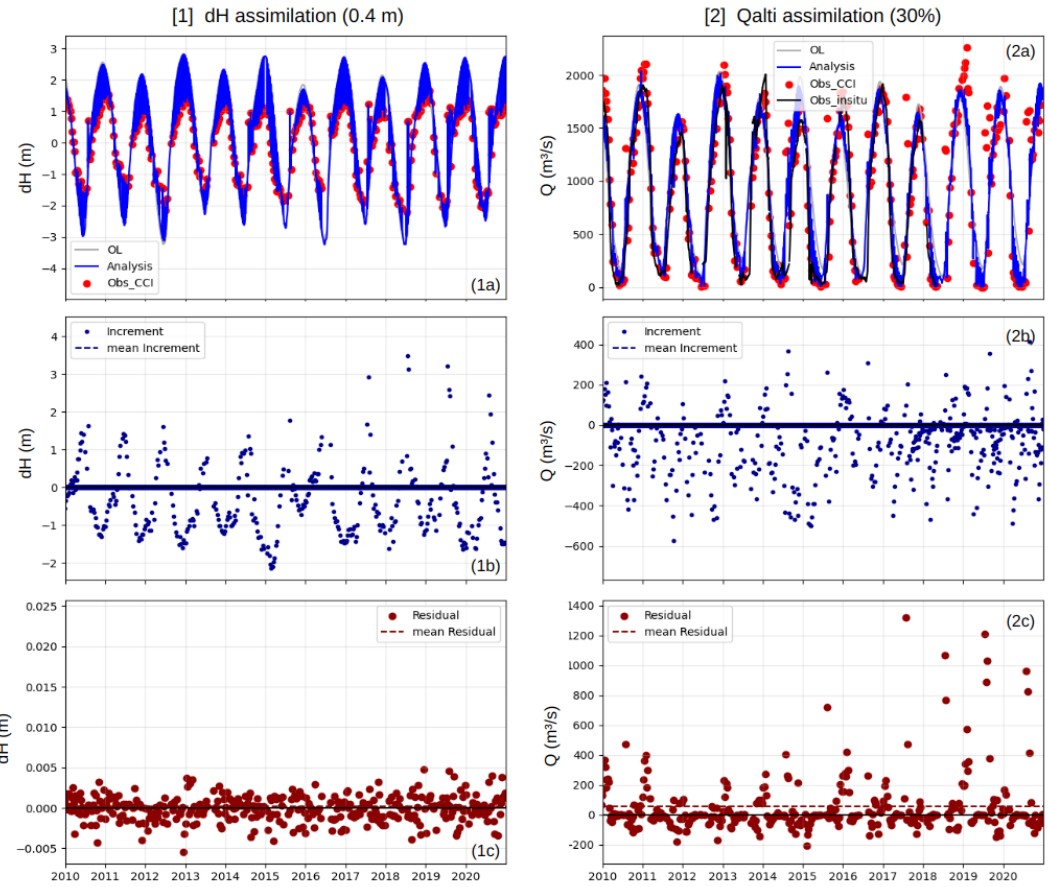

**Figure 19.** Comparison of dH and Q assimilation at Ansongo, Niger, over the 2010-2020 period: Top panels show time series of (a) dH assimilation with 0.4m uncertainty and (b) Qalti assimilation with 30% uncertainty, with open-loop, analysis, and observed data. The middle and lower panels show the corresponding innovation and residuals, with the mean innovation and residuals represented by dashed lines, illustrating the model's performance and adjustments after assimilation.

with the observed altimetry data, as seen in Figure 22, allowing for near-total dH observation assimilation compared to Qalti, despite both products being derived from the same altimetry source (Figure 22). The lower uncertainty of dH plays a crucial role in this case, as it drives better assimilation outcomes by assigning more weight to the observed water surface elevations, resulting in smoother corrections to the model's discharge predictions.

The preferred assimilation variable depends on the model's open-loop performance, the uncertainty assigned to each variable, and how they are represented in the model. The issues with rating curve mismatches in MGB (Kinshasa, Figure 20) and the similar performance results between dH and Qalti in CTRIP (Figure 22) demonstrate that, overall, discharge (Q) remains a more effective variable for assimilation in large-scale models. These findings underscore that despite some close comparisons,





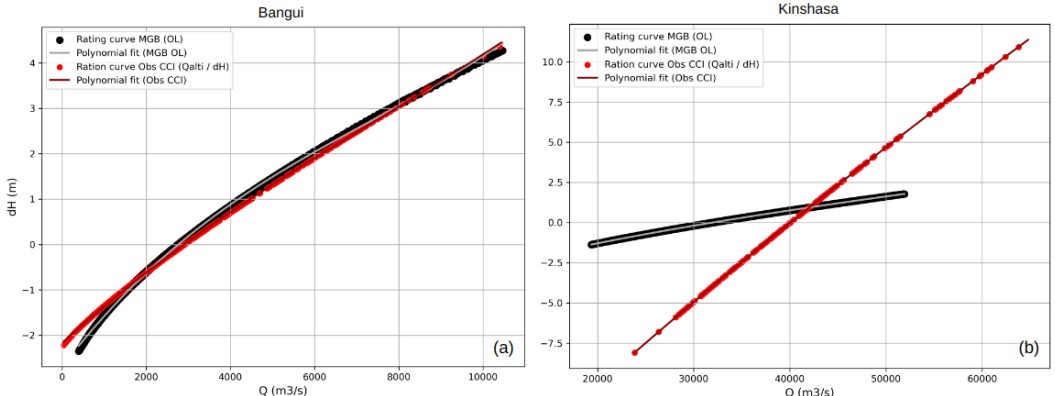

**Figure 20.** Rating curves from MGB (Open-Loop) vs. Obs CCI (dH, Qalti) at Bangui and Kinshasa stations in the Congo basin.

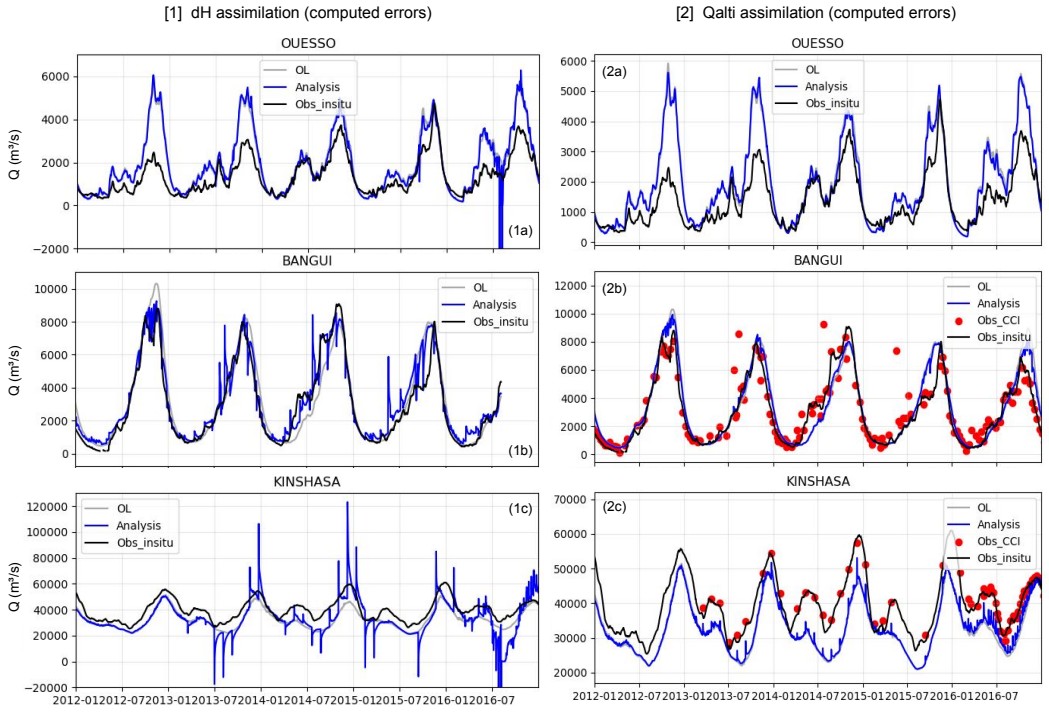

**Figure 21.** Comparison of dH (1) and Qalti (2) Assimilation in Ouesso, Bangui, and Kinshasa: OL Simulation in black, Assimilation Results in blue and Observed CCIP Data in red.

discharge offers a more reliable and direct integration into hydrological models, avoiding the complexities associated with converting water surface elevation (dH) to discharge.





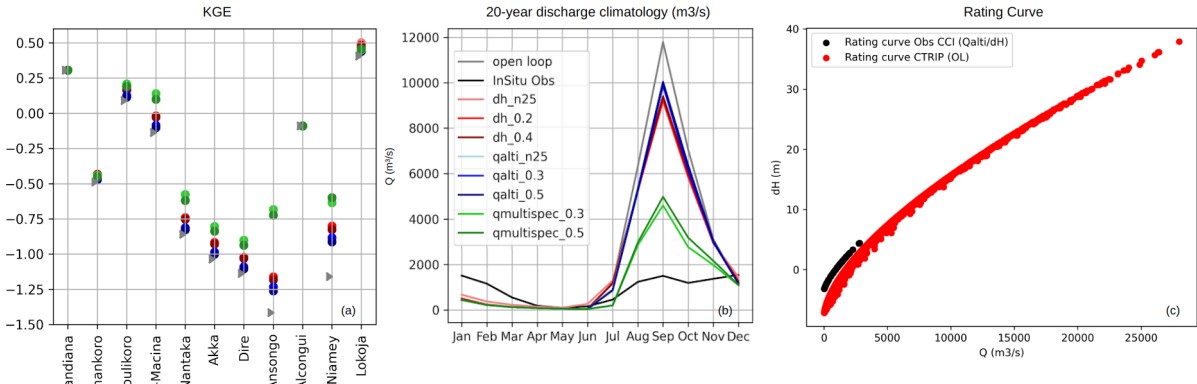

**Figure 22.** Example of Niamey, rating curve and climatology

For large-scale models, whose primary objective is to accurately simulate discharge rather than water surface elevation (WSE), it is more robust to assimilate a variable directly related to the flows they simulate. Adjusting rating curves across model cells to accommodate dH is more complex and introduces errors, especially when the model's simplified river geometry

does not match real-world conditions. Discharge integrates smoothly into models like MGB and CTRIP without requiring these complex conversions, making it the more reliable choice overall.

### 4.2.3 Bias and internal variability correction : Outperformance of Qmultispec assimilation over Qalti

In the Niger Basin, despite its lower performance in other metrics, Qmultispec outperforms in bias correction by significantly reducing flow overestimation in both models. This improvement is particularly notable in the upper Niger and Inner Delta

within MGB, where the Open-Loop (OL) simulation consistently overestimates discharge. Koulikoro serves as a key example, where the bias is reduced by 17-20% (depending on the uncertainty levels), bringing the bias from 1.2 in the Open-Loop down to approximately 1. In fact, Qmultispec's higher temporal density enables better alignment with the observed rising and falling limbs of the hydrograph, providing more consistent results over the annual cycle.

Figure 23 illustrates in panels (a) the time series results for two experiments conducted at Koulikoro between 2010 and

2020 (MGB model): (1) Qalti and (2) Qmultispec assimilation, both at 30% uncertainty. In panels (b), the time series of increment – which measures the difference between the analysis and the background – for each experiment are shown (Qalti in 1b and Qmultispec in 2b). We observe that with frequent and continuous data input, the bias correction occurs steadily over the annual cycles. Both the overestimation during high-flow periods and the underestimation during low-flow periods are corrected consistently.

As shown in panel b of Figure 24, the increment displays a notable reduction during the high-flow period (June to November), with a much lower mean increment for Qmultispec than for WSE (dH) or Qalti assimilation. At Koulikoro, Qalti actually overestimates discharge by 35% compared to in-situ data, while the MGB Open-Loop overestimates by 25%, and Qmultispec





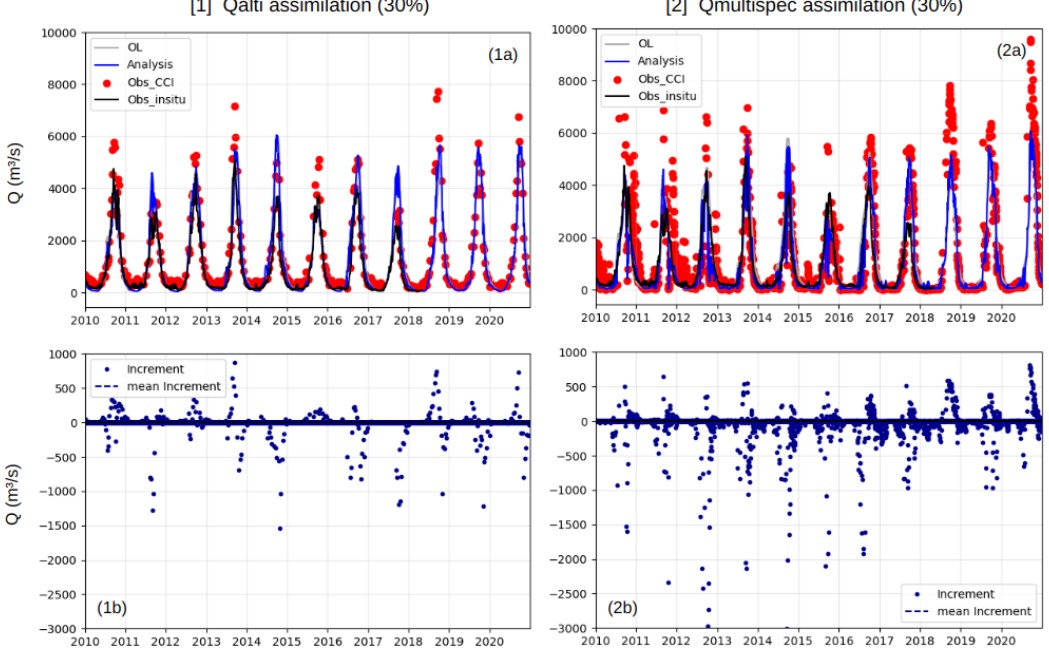

**Figure 23.** MGB model results at Koulikoro over 2010-2020 for two experiments: (1) Qalti with 30% uncertainty and (2) Qmultispec with 30% uncertainty. Time series for open-loop (black), Analysis (blue), and Observations to be assimilated (red) are shown in panels (1a) and (2a) for Qalti and Qmultispec, respectively. The increment (Analysis - Background) is shown in (b), and the residual (Obs CCI - Analysis) in (c). Dashed lines in (b) and (c) represent the mean innovation and residual over 20 years

by 20%. Although the Qmultispec data is noisier, particularly during the recession phase, its high temporal resolution allows for continuous correction, preventing the model from reverting to its pre-assimilation, overestimated state. In panel (2b) of

Figure 23, showing the increment over time, we see that updates occur continuously with a repetitive trend throughout the annual cycle.

In the CTRIP model, the order of improvement follows a similar trend, with Qmultispec providing the greatest reduction in bias, followed by WSE anomalies (dH) and altimetry-based discharge (Qalti). This difference stems from the lower uncertainty levels of WSE anomalies (0.2m-0.4m) compared to Qalti (30%-50%). In the Niger Basin, CTRIP's Open-Loop simulation

overestimates discharge significantly, so all satellite products help reduce this positive bias. The higher temporal resolution of Qmultispec prevents the model from reverting to its pre-assimilation state, reducing bias more effectively than the other products. This impact is particularly evident in the averaged monthly discharge values, where Qmultispec consistently shows better bias reduction than both WSE anomalies and Qalti assimilation, as seen in Niamey (Figure 22).

The dense temporal coverage of Qmultispec also ensures better capture of internal variability and long-term trends, more

seen in MGB model results. Discharge data with higher temporal resolution provides a more dynamic correction, reducing the divergences in model simulations caused by internal errors or the omission of complex physical processes. This is evident in



**Figure 24.** (a, b) MGB model results at Koulikoro for different experiments (dH in shades of red, Qalti in shades of blue, Qmultispec in shades of green); (a) Discharge climatologies, with in-situ observations in black. (b) Increment (Analysis minus Background) shows differences after assimilation, with mean increments indicated by dashed lines. (c, d) Example of assimilation results over a year (2014) for Qalti (blue) vs Qmultispec (green) at Koulikoro within MGB (c) and CTRIP (d), both CCI products assimilated at 30% uncertainty.





both the MGB and CTRIP models, as seen in Figure 24. It is worth noting that within the Niger basin, the internal variability is less effectively corrected by any of the products in CTRIP, largely due to the model's poor Open-Loop performance, which limits its ability to capture short-term fluctuations.

The outperformance of Qmultispec in internal variability correction compared to Qalti and WSE, makes this product particularly valuable for climate studies focusing on flow variability and its evolution over time. Capturing this internal short-term variability is essential for modeling extreme events such as floods and droughts, which makes this product doubly relevant for MGB, as it is designed for forecasting applications.

Overall, in both models, Qmultispec stands out as the most effective product for reducing bias and correcting internal variability. In particular, its higher temporal resolution provides continuous corrections to both high- and low-flow periods, reducing the existing biases in Open-Loop simulations. Despite the noisiness in the data, Qmultispec delivers more consistent and reliable performance compared to Qalti and WSE anomalies. This makes it particularly valuable for long-term studies on the evolution of discharge and water resources, such as those related to climate change, as well as for forecasting extreme events.

## 4.3  Discussions

This study demonstrates that while discharge data assimilation yields more reliable results, especially in discharge-driven models, the use of WSE anomalies can introduce significant errors. For instance, in Kinshasa, the discrepancy between the modeled and observed WSE values resulted in poor model performance when assimilating dH, due to mismatches between the model's rating curve and real-world conditions. Conversely, assimilation of discharge aligns better with the modeled variables, offering smoother and more consistent corrections, which are particularly vital for simulating discharge patterns accurately. This underlines the need for careful consideration when integrating dH data into discharge models, especially in cases where the river bathymetry and geometry are not well represented.

Temporal data density plays a critical role in hydrological modeling, particularly for climate studies where capturing internal variability and correcting biases are essential. High-frequency data, like Qmultispec, provides continuous updates that help correct seasonal and interannual flow patterns, which are important for modeling extreme events such as floods and droughts. In the Niger Basin, Qmultispec assimilation significantly improved the model's representation of seasonal flows and reduced systematic biases. However, a trade-off exists between data quality and temporal density. High-frequency datasets, such as Qmultispec, tend to be noisier compared to lower-frequency, but more accurate datasets like Qalti. In regions such as Bangui in the Congo Basin, where Qmultispec data was noisier, the assimilation led to poor model performance due to the excessive noise, despite its high temporal resolution. This highlights the importance of balancing the density of observations with their quality to avoid degrading the model with inaccurate data. Figure 25 illustrates this trade-off, with high-frequency data improving flow timing but introducing noise into the analysis.

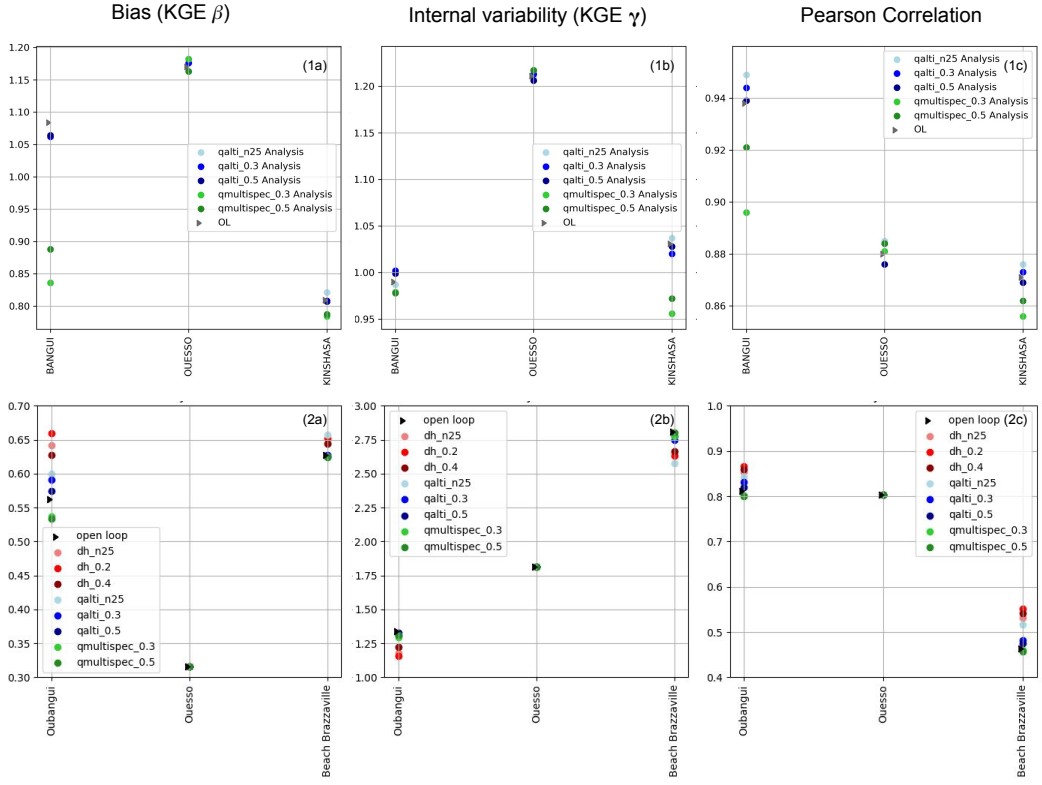

**Figure 25.** Comparison of bias (KGE $\beta$), internal variability (KGE $\gamma$), and Pearson correlation for different assimilation experiments in MGB and CTRIP models: daily scores at Ouesso, Bangui, and Kinshasa stations.

The trade-off between temporal density and data quality is especially evident in the Congo Basin, where Qmultispec performs poorly compared to its performance in the Niger Basin. In the Congo, the lower data quality and poorer temporal density

reduce the effectiveness of Qmultispec, particularly in stations like Bangui and Kinshasa. While CTRIP is more influenced by WSE and discharge assimilation, the mismatches between the modeled and observed rating curves in MGB create additional challenges, leading to errors when assimilating dH. Furthermore, at stations like Ouesso, the high-water periods in Qmultispec are out of phase with the modeled output, diminishing the impact of assimilation.

Spatial data density is another critical factor in complex river basins like the Congo. The Congo's major tributaries con-

630 tribute significantly to the overall discharge, necessitating adequate observation points for proper basin-wide hydrological representation. For example, insufficient coverage in the upper tributaries of the Congo, such as the Chembe-ferry station, limits the overall influence of data assimilation across the basin. Figure 26 presents the normalized root mean square difference (NRMSD) between the Open-Loop and Qalti (0.20m) assimilation experiments over both basins. As shown in this figure, the impact of spatial coverage on data assimilation effectiveness is significant, especially when comparing the spatial coverage in

the Niger Basin to the Congo Basin.





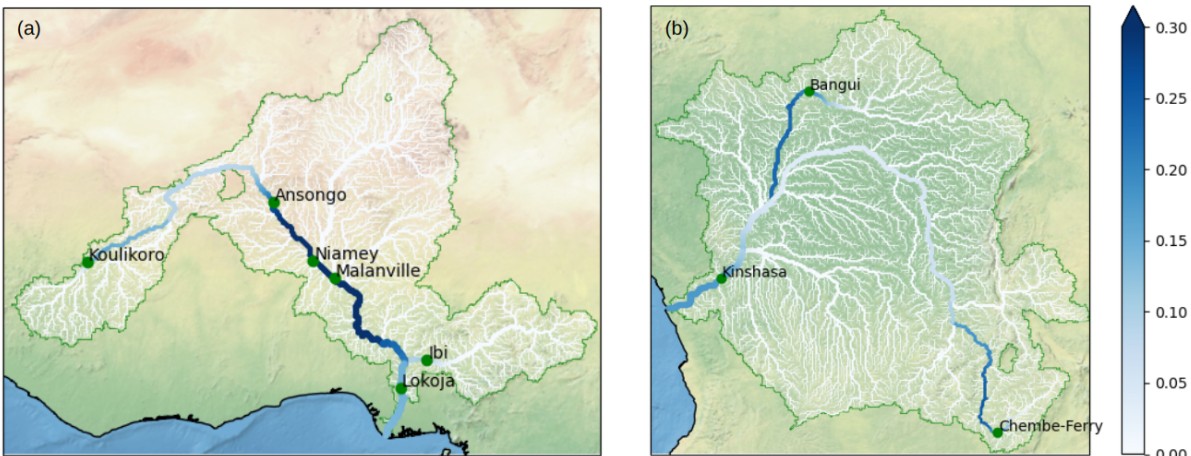

**Figure 26.** NRSMD comparison between Niger and Congo basins, illustrating the impact of spatial coverage on DA influence.

In summary, the effectiveness of data assimilation in large-scale hydrological models depends on several key factors: the type of data assimilated (discharge vs. WSE), the temporal and spatial density of observations, and the initial performance of the model. While discharge assimilation proves to be more reliable in models designed for flow dynamics, the quality of the observation data, particularly in complex basins like the Congo, plays a crucial role in determining the success of assimilation experiments.

## 5 Conclusions and Perspectives

Hydrological models such as CTRIP and MGB are essential tools for climate studies and forecasting at large scales, including continental regions and large river basins. Their accuracy can be greatly enhanced through satellite data assimilation, which improves model performance by addressing key factors such as the model's initial open-loop performance, the temporal and spatial density of observations, and the quality of the assimilated data. Assimilating discharge data has proven more reliable than water surface elevation anomalies (dH) assimilation, particularly in models focused on discharge simulations. This is because direct discharge assimilation avoids the uncertainties involved in converting dH into discharge, especially in models with simplified river geometry. In contrast, dH assimilation often introduces errors in regions where discrepancies exist between observed and modeled water surface elevations.

Temporal data density is also critical for improving model performance. High-frequency updates, such as those provided by Qmultispec, enable more continuous corrections, reducing bias and better capturing flow variability over time. However, noisy data can reduce model performance if not balanced with data quality, as seen in the Congo Basin at Bangui, where Qmultispec's higher temporal resolution introduced noise into the model. High-quality data like Qalti, which has lower uncertainty, consistently reduces bias and improves model accuracy, though its limited availability restricts the frequency of model updates.

Spatial data density is equally important, particularly in complex river basins like the Congo, where major tributaries play a





crucial role in the overall discharge. Insufficient spatial coverage can result in errors in basin-wide simulations, whereas better station coverage in regions like the Niger Basin has led to more consistent improvements.

Looking ahead, Phase 2 of the CCI Discharge project presents significant opportunities for improvement. The focus will shift to assessing how enhanced space-time sampling and uncertainty quantification influence the performance of data assimilation. This phase will also incorporate outlier detection and combined data assimilation methods for both discharge and dH in order to densify the spatial coverage. By leveraging the high spatial coverage of dH and the direct relevance of Q, these combined approaches are expected to enhance models accuracy like CTRIP, especially in regions with low observation density. or the MGB model, challenges may arise with certain rating curves, making dH less suitable for assimilation. However, in areas where simulated and observed rating curves align, the combined assimilation of these datasets could prove beneficial. Additional experiments will be conducted using newly developed datasets for large scale hydrological models, CTRIP and MGB, assessing the impact of combined assimilation on hydrological simulation.

Furthermore, the integration of SWOT (Surface Water and Ocean Topography) satellite data will offers distinct advantages for enhancing models accuracy through its detailed measurements of water surface slope, further reducing uncertainties in discharge predictions. These advantages, combined with the previously discussed improvements in space-time sampling and refined uncertainty quantification, will enable models to better represent critical parameters such as slope. This synergy is expected to significantly reduce uncertainties in discharge predictions and improve the overall performance of the models in simulating river discharge and water storage dynamics.

Future improvements in data assimilation schemes will be crucial for further enhancing model performance, with one promising approach being the implementation of dynamic localization distances in models like CTRIP. This innovation will enable models to adapt to both natural variability and anthropogenic influences, such as the effects of dams and irrigation on river systems. By incorporating these dynamic localization techniques, models will be better equipped to account for changes in hydrological processes, thereby improving the reliability of hydrological forecasts.

As models continue to evolve, namely in improving parameterization or in integrating new processes (like evaporation over the Niger Inner Delta for the CTRIP model), the focus will be on refining data assimilation schemes to maximize the benefits of satellite data. The introduction of smoother methods, especially in Kalman filter-based systems, could mitigate the hashed effects observed in dH assimilation, ensuring more accurate corrections. These advancements, combined with SWOT data integration, will address current gaps in data coverage and enhance the simulation of climate-impacted and human-modified river systems.

*Data availability.* The CCI-Discharge datasets are available at the following link: https://climate.esa.int/en/projects/river-discharge/ ("Data" section). The Product User Guides for each product are available in the "Key Document" section.



*Author contributions.* MS, GN, VP, and SM conceived the research design and experimental setup. MS and GN processed the data, ran the experiments, processed the results, and analyzed the findings. VP, SM, and KV assisted with data assimilation tools setup (SM and KV developed/improved HyDAS for CTRIP, and VP developed HYFAA in MGB) and contributed to the discussions. MS merged the findings of both models and wrote the initial draft, with input from GN, SM, and VP. CA, SB, and AA reviewed the paper and provided valuable 690 insights. All authors contributed to the final version of the paper.

*Competing interests.* The contact author has declared that none of the authors has any competing interests.

*Acknowledgements.* We acknowledge all ESA Climate Change Initiative (CCI) project members for producing and providing the water surface elevation (WSE) and discharge products (Qalti, Qmultispec) used in this study. We also thank Adrien Paris and Laetitia Gal from *Hydromatters* compagny for the MGB model setup over the Niger and Congo basins, as well as the CNRM and the French Space Agency 695 (CNES) for providing the necessary computing infrastructure. Additionally, we appreciate the Niger Basin Agency (ABN) and So-HyBam Environmntal and Research project for making in-situ discharge and WSE data available over the Niger and Congo basins, which were essential to analyze our results.





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
