# Peer review of "Improving large-scale river routing models for climate studies: the impact of ESA long-term CCI discharge products on correcting multi-model hydrological simulations"

_Hydrology and Earth System Sciences, 2024_

## Author Comment (AC1)

**Responses to Comments of Reviewer #1**

**Major Comments:**

1. I strongly recommend revising the **title of the manuscript**. It needs to reflect the content of the manuscript. For example, "correcting" is too broad as the manuscript is about assimilating data into a hydrological/hydrodynamic model. In addition, even though the authors mention "climate studies" in the title, the authors seldom discuss how these methods be used in climate studies.

    Response :

We appreciate the reviewer's suggestion to revise the title to better reflect the content of the manuscript. We agree that "correcting" is too broad and that the connection to "climate studies" should be made more explicit, both in the title and throughout the manuscript.

Our study focuses on demonstrating how assimilating long-term, daily data series from the ESA Climate Change Initiative (CCI) can enhance large-scale hydrological models by reducing cumulative biases and improving flow variability representation. These improvements are critical for ensuring that models provide more reliable long-term hydrological simulations, which are essential for climate studies—particularly for understanding trends and variability over decades.

We acknowledge that the current introduction (lines 43–77) does not sufficiently clarify the broader climate context of this study. While we do not conduct a climate study per se, these long-term discharge products and our work on analyzing their assimilation into large-scale hydrological models represent an initial and promising step toward further climate applications, such as the production of hydrological reanalyses. One of the key advantages of assimilating long-term datasets, such as CCI discharge, is ensuring consistency over multi-decadal periods, thereby avoiding artificial trends that could arise when using only recent satellite missions. Ideally, climate studies require at least 30 years of consistent data, and while the current CCI dataset spans 20 years, it represents a crucial step toward that goal.

To clarify these points, we will revise the introduction to better position this study within the broader objective of long-term hydrological modeling and climate-related applications. We will explicitly state that while this paper does not perform a climate analysis, its findings support improvements in large-scale hydrological models that are essential for future climate impact studies.

To better reflect these objectives, we propose the following revised title:

"Improving Large-Scale River Routing Models with ESA long-term CCI discharge data assimilation: Advancing Accuracy for Climate Studies."

2. The description of the data assimilation ("2.3.2 The Ensemble Kalman Filter (EnKF)") is more focused on the data assimilation method used in CTRIP-HyDAS despite the manuscript addressing both CTRIP-HyDAS and MGB-HYFAA.
    a. The authors should include an explanation of the methods used in MGB-HYFAA. Is MGB-HYFAA based on Wongchuig-Correa et al., (2020); Wongchuig et al., (2023, 2024). It appears that MGB-HYFAA uses the Local Ensemble Kalman Filter (LEnKF), whereas CTRIP-HyDAS employs the Local Ensemble Transformation Kalman Filter (LETKF).
    b. Additionally, it is unclear whether the same localization approach is used in data assimilation frameworks. The authors should clarify whether CTRIP-HyDAS and MGB-HYFAA apply identical observation localization methods.
    c. Furthermore, the differences between CTRIP-HyDAS and MGB-HYFAA in the context of data assimilation should be discussed. Summarizing the key features of HyDAS and HYFAA—such as routing schemes, data assimilation methods, and localization approaches—would greatly improve clarity.

Response :

We appreciate the reviewer's insightful comments regarding the data assimilation (DA) methods employed in our study.

Both CTRIP-HyDAS and MGB-HYFAA employ ensemble-based Kalman Filter methods, but there are notable differences in their algorithms: CTRIP-HyDAS uses the Local Ensemble Transform Kalman Filter (LETKF), which updates the ensemble mean and perturbations separately, ensuring that the updated ensemble maintains the correct statistical properties; while MGB-HYFAA employs Local Ensemble Kalman Filter (LEnKF), which updates each ensemble member directly based on the observations and the model's error statistics. We agree that these distinctions must be explained.

For the localisation technique, both DA schemes incorporate the same algorithm from Revel et al. (2019) to limit the spatial influence of observations over parts of the river network that are hydrodynamically consistent. In both schemes, the localisation of each pixel depends on the hydrological connectivity within the basin, based on physically based distances from semi-variogram of a 20-year long-term free run of the models (2000-2020).

To enhance clarity, we propose to modify subsections 2.3.2 and 2.3.3 as follows :

- Overview of the Ensemble Kalman Filter (EnKF), detailing its general principles and equations.
- Specify the implementations within HYDAS and HYFAA, highlighting the distinctions between LETKF used by CTRIP-HyDAS and LEnKF in MGB-HYFAA.
- As both schemes incorporate the same localization technique from Revel et al. (2019), we will retain the description used in lines 216-222 and make it explicit that both CTRIP-HyDAS and MGB-HYFAA utilize this approach.

- Comparative table as a conclusion for the section, summarizing the key features of CTRIP-HyDAS and MGB-HYFAA, including information about the forcings used, their routing schemes, data assimilation methods, and localization approach, with the respective references.

We believe these revisions will provide readers with a clearer understanding of the distinct DA frameworks and their respective methodologies.

3. The manuscript is overly lengthy and contains 26 figures, which is excessive for a research article. To improve readability and focus, the authors are encouraged to condense the content and move some sections to supplementary materials or an appendix. For instance, the evaluation of satellite-based discharges and the open-loop performance of the models could be relocated to supplementary materials. This would streamline the main text, ensuring that the most critical findings and discussions remain central, while detailed analyses and additional data are still accessible for interested readers.

Response :

We thank the reviewer for feedback regarding the manuscript's length and the number of figures. To enhance readability and maintain focus on the central findings, we agree with the proposition of relocating Sections 2.4.2 and 4.1 (excluding Subsection 4.1.3) to the supplementary materials so that the detailed evaluations of satellite-based discharges and the open-loop performance of the models remain accessible without diverting attention from the main narrative. The five figures associated with these sections will also be moved to the supplementary materials.

We will retain Subsection 4.1.3 in the main manuscript, as it synthesizes key issues in both models, providing context for interpreting the data assimilation results and offering initial insights into expected improvements. To ensure coherence, we will introduce this subsection with a brief summary of the hydrological models' open-loop response.

Figure 8 will be kept in the main manuscript (to be positioned after Table 1) and be included in the supplementary materials, as we find the maps important for readers to understand the spatial distribution of available observations across each basin.

We will also condense the figures in response to the remaining comments from both reviewers.

4. Based on the explanation in Section 2.4.1, Table 2, and Figure 8, it appears that the authors have used only satellite observations corresponding to GRDC locations. Why didn't the authors utilize all the available satellite altimetry data across the entire basin? The strength of satellite data lies in its spatial distribution, even if it is limited in temporal resolution. While Qalti and Qmultisepc generate discharge data only at locations where in-situ observations are available, WSE has the potential to provide coverage across the entire basin. To fully exploit the potential of satellite altimetry, the authors could incorporate all available satellite data across the basin.

Response :

Our response to Comment 4 is closely linked to Comment 1, highlighting the need to clarify the study's objectives throughout the manuscript. This study is specifically designed to assess the added value of long-term (20 years), high-temporal resolution (near-daily) Water Surface Elevation (WSE) and discharge products in improving large-scale hydrological models. The goal is to evaluate how these datasets contribute to long-term hydrological modeling in the context of climate studies.

While integrating additional satellite data with different spatial and temporal resolutions could potentially improve model accuracy, it would also make it difficult to isolate the specific contributions of the CCI products. Our aim is not to optimize data assimilation performance by maximizing observation density but rather to quantify the distinct benefits of long-term discharge and WSE datasets. Introducing additional data sources would add factors such as spatial density and observation frequency, making it harder to determine whether improvements stem from the long-term consistency of the CCI products or from increased observational coverage. To maintain a clear focus on the standalone impact of long-term data, we have deliberately chosen to rely solely on CCI products in this proof-of-concept study.

Another key aspect of this study is the comparison between the assimilation of long-term discharge data and WSE data, focusing on how each contributes to hydrological model performance. While a denser observational network would likely improve results, our approach aims to evaluate whether long-term discharge data provide specific advantages over WSE for improving large-scale hydrological models.

We acknowledge the potential advantages of integrating all available satellite data to enhance spatial coverage and model accuracy. This is something we plan to explore in the next phase of our research (CCI Discharge Phase 2, initiated in November 2024), where we will investigate the impact of integrating the CCI products with other available satellite observations to better understand their combined potential in data assimilation in enhancing hydrological model performance. This aligns with discussions in our manuscript regarding the limitations and potential extensions. By building on the insights gained here, we aim to explore how multi-source satellite data can further improve large-scale hydrological modeling.

Finally, this comment underscores that the study's objectives and context were not made sufficiently explicit in the introduction and throughout the manuscript. As noted in our response to Comment 1, we will revise the introduction (lines 43–77) to better explain the study's scope and the rationale behind using only the long-term CCI products. We will also refine the discussion section to emphasize that this study serves as an initial step toward broader climate applications, such as hydrological reanalyses and climate impact assessments.

5. Furthermore, the authors could explore and discuss the added value of spatially distributed observations for assimilation. Relying solely on satellite data at GRDC gauges may fail to leverage the advantages of data assimilation localization (as noted in Line 637), particularly since no additional observations would influence the assimilation in nearby river reaches.

Response :

In our current study, we focused on assimilating satellite-derived data at GRDC gauge locations primarily because these sites offer the possibility to merge satellite data from multiple missions (with inter-mission bias removal) and then generate long-term records (possibly longer than those from in-situ gauges), which are essential for our analysis. We recognize that relying solely on data from GRDC gauges may limit the effectiveness of data assimilation localization techniques, as it restricts the influence of observations on nearby river reaches. The rest of the spatially distributed observations have limited temporal coverage, posing challenges for studies like ours that require long-term data to assess hydrological trends and model performance over extended periods. The multispectral discharge product, a new type of discharge data, relies on multispectral images instead of in-situ gauge stations. Due to the longevity of the records and the abundance of multispectral images, this product offers extended coverage of 20 years with higher temporal density.

In future research, we plan to incorporate additional spatially distributed observations, especially in regions lacking in-situ stations, to enhance the localization process and improve model performance. We will emphasize these future directions in the manuscript's perspectives section.

6. The hashed effect observed in the assimilated products seems to result from the sparseness of observations and the use of observation localization. From the description in Lines 504–512 and Figure 17, it is unclear whether all observations within the 'local patch' (as shown in Figure 7) are utilized. It would be valuable to investigate how many observations can be accommodated within a single local patch. If the local patch is large enough to include a substantial number of observations, the hashed effect could be minimized. Therefore, the authors are encouraged to analyze the effect of local patch size and the number of observations within each patch to determine their impact on the results.

Response :

We appreciate the reviewer's suggestion to examine how the size of local patches and the number of observations within them might influence the "hashed" effect observed in our data assimilation process. Upon analysis, we believe this effect is not directly related to the size of the local patch. While increasing the number of observations within a local patch could provide some correction along the river reach, the automatic reversion to the open-loop state observed after each assimilation step indicates a significant discrepancy between the assimilated Water Surface Elevation (WSE) observations and the model simulations.  These results suggest that the model dynamics is quite fast, limiting the persistence of the assimilation. Besides, it has to be noted that using the localization allows to correct the model state upstream the observed pixel and this correction then propagates downstream on the following time steps, which may indirectly increase the assimilation persistence.

We have observed that this "hashed" effect is more pronounced when assimilating WSE data compared to discharge data. This observation leads us to hypothesize that the primary cause of the effect is the mismatch between the WSE observations and the simulated states, rather than the configuration of the local patch. In contrast, discharge assimilation shows a more consistent adjustment.

When developing our methodology, we conducted exploratory tests in the Niger basin using MGB-HYFAA with three localization strategies: no localization, constant-distance localization (300 km), and adaptive localization based on semi-variograms computed for each pixel. Adaptive localization generally reduced the hashed effect compared to the other approaches by better accounting for hydrological connectivity (as explained previously), but the hashed effect remains present after each assimilation in stations where there is a big misalignment between simulated and observed rating curves and where these WSE are available and assimilated.

Related to this point, as mentioned in our manuscript, future work will focus on developing smoother methods to evaluate if they help dampen the hashed effect, particularly the remaining effect when assimilating discharge. These approaches could help refine the corrections applied during data assimilation. However, implementing this method is beyond the scope of this current study, as they require further methodological advancements to be integrated into MGB-HYFAA.

7. The authors have generally stated that discharge is the preferred variable to assimilate, citing mismatches in the rating curves as a reason. However, this explanation lacks depth. The authors should delve deeper into the specific circumstances under which discharge is a more suitable variable for assimilation. They should provide a more detailed analysis, outlining the advantages and limitations of assimilating discharge versus other variables such as water surface elevation (WSE), particularly in the context of large-scale river models. Moreover, offering guidance on how to select the most appropriate variables for data assimilation would significantly enhance the manuscript. This guidance could include considerations such as data availability, model structure, observation accuracy, and the specific objectives of the assimilation. Such an exploration would not only strengthen the current study but also serve as a valuable resource for researchers working with large-scale river models.

   Response :

We acknowledge the reviewer's request to explore the specific circumstances under which discharge is a more suitable variable for assimilation compared to WSE. This is indeed an important question, and while we provide some interpretations and explanations in our study, we acknowledge that the results are highly dependent on the model used and potentially on data quality.

In this study, we find that discharge assimilation generally leads to better performance, though not in all cases (this also addresses moderate comment #10). We discuss the

advantages of discharge assimilation, particularly its alignment with hydrological modeling objectives and its reduced dependence on rating curve transformations.

To strengthen Section 4.2.2 and provide additional context, we propose adding the following explanations:

- Both models used in this study, MGB and CTRIP, are large-scale hydrological models primarily designed for streamflow prediction in applications such as water resource management and flood forecasting. These models simplify river representation using rectangular channels, and flow dynamics are computed with Manning-Strickler equations. Since their primary output is discharge, which is later converted into WSE through estimated rating curves where available, discharge is inherently a more relevant variable for assimilation.
- Assimilating discharge presents two key advantages: (1) it directly aligns with the objectives of hydrological models, and (2) it reduces dependence on rating curves, which can introduce significant errors. For instance, at stations like Ansongo and Kinshasa in the Niger and Congo basins, respectively, mismatches between simulated and observed rating curves lead to notable discrepancies in WSE assimilation (Figures 18, 20). By assimilating discharge, these issues are mitigated, leading to more robust model corrections.
- However, the simplified equations and structural assumptions in both models also influence how effectively they simulate hydrological processes. For example, CTRIP does not fully represent evaporation dynamics in the inner delta of the Niger, leading to discrepancies between observed and simulated open-loop discharge, which in turn affects assimilation outcomes. In contrast, MGB, which explicitly simulates evapotranspiration and has undergone calibration, better captures these dynamics.
- Another advantage of discharge assimilation lies in its temporal continuity. The long-term CCI discharge products provide a consistent, nearly-daily time series, allowing the models to better capture temporal variability in water volumes. This is evident at stations like Malanville (Panels 1c-d vs. 2c-d in Figure 18), where well-aligned rating curves and a sufficiently broad local patch result in improved agreement between observed and simulated streamflow after discharge assimilation. Similarly, for Qmultispec products, which have a higher temporal density, the reduction of bias and improved internal variability further emphasize the benefits of discharge assimilation.

[Figure]

*Localisation for Malanville based on adaptive distance (semi-variogramm) (Niger Basin)*

While we acknowledge the importance of further analyzing the choice between discharge and WSE assimilation, given the already extensive scope of this manuscript, we cannot comprehensively address all aspects. We have, however, already conducted sensitivity tests on observation uncertainties, which demonstrated that model performance is more sensitive to the assimilated variable than to the assigned observation errors (Figures 15–16). Additionally, we analyzed how rating curve discrepancies impact WSE assimilation, particularly in relation to the hashed effect.

We therefore propose several avenues that could be explored in future work to deepen this analysis. For instance, investigating the behavior of routing reservoirs in MGB or relevant RRM variables in CTRIP during assimilation could help understand how each model reacts to different assimilated variables. A seasonal analysis of assimilation effectiveness could further highlight the relative performance of discharge versus WSE during low-flow and high-flow periods. Additionally, incorporating additional remote sensing data, such as river width and slope (e.g., from SWOT), into the assimilation framework could allow for simultaneous state and parameter updates, offering a more comprehensive approach. However, these aspects go beyond the objectives of the present study and are suggested as perspectives for future work.

We hope this clarification helps put our findings into perspective within the broader discussion on discharge versus WSE assimilation in large-scale hydrological models, while also highlighting the need for further research on this topic.

8. The manuscript does not address the impact of the forcing data on the final assimilation results. CTRIP uses ERA5 forcing data, while MGB employs GSMAP/IMERG and ERA5. Given that precipitation may differ between ERA5 and GSMAP/IMERG, it is important to assess how this discrepancy influences the assimilation performance. If this analysis is beyond the scope of the manuscript, the authors should consider using the same forcing data for both models to ensure a fair comparison.

Response :

We thank the reviewer for highlighting the importance of assessing the impact of forcing data on assimilation results. However, it is important to clarify that CTRIP and MGB are existing, ready-to-use modeling chains, each designed for distinct applications. CTRIP is tailored for climate studies, water resource management, and long-term hydrological projections, while MGB is primarily used for operational forecasting over large river basins. The differences in forcing data reflect the specific design and operational goals of these models.

For CTRIP, ERA5 forcing data was chosen for its global coverage, temporal resolution and consistency, with its intended use for climate studies and water resource assessments (Dee et al., 2011; Hersbach et al., 2020). Additionally, efforts have been made to bias-correct ERA5 precipitation to improve its accuracy in the hydrological context. This bias-correction process is outlined in (Szczypta et al., 2012).

For MGB, the model addresses a significant limitation with the ERA5 precipitation product. In fact, existing biases in ERA5 precipitation data were found to significantly degrade model performance in operational forecasting. GSMAP and CHIRPS were selected due to their higher spatial resolution and temporal frequency, which better capture localized rainfall events in the Niger and Congo basins, which better align with MGB's forecasting objectives. For the Niger Basin, GSMaP provides near-real-time global precipitation data with high temporal resolution (hourly) and spatial resolution (0.1°). It has been shown to perform well in tropical regions, as highlighted by Kubota et al. (2020). For the Congo Basin, CHIRPS was chosen due to its ability to integrate satellite imagery with in-situ observations, providing robust long-term rainfall data with a resolution of 0.05° and daily temporal resolution. Its performance in the Congo Basin is detailed in studies by Gosset et al. (2023) and Peterson et al. (2015).

In the revised manuscript, we will provide these additional details about the choice of forcing data for both models and include the references above to clarify their relevance to the respective basins in Section 2 when presenting the CTRIP and MGB models.

We acknowledge the reviewer's suggestion to use the same forcing data for both models to ensure a fair comparison. However, given the specific contexts of development of CTRIP and MGB chains and their operational objectives, this analysis is beyond the scope of this study. Instead, our focus is on evaluating how the assimilation of long-term satellite-derived observations improves model performance within established modeling chains. While using the same forcing data for both models could allow for a more direct comparison, such an adjustment would not reflect the independent operational setups and study objectives of these two chains. We will note this limitation explicitly in the revised text to ensure transparency.

References:

Dee, D. P., Uppala, S. M., Simmons, A. J., Berrisford, P., Poli, P., Kobayashi, S., Andrae, U., Balmaseda, M. A., Balsamo, G., Bauer, P., Bechtold, P., Beljaars, A. C. M., Van De

Berg, L., Bidlot, J., Bormann, N., Delsol, C., Dragani, R., Fuentes, M., Geer, A. J., … Vitart, F. (2011). The ERA‑Interim reanalysis: Configuration and performance of the data assimilation system. Quarterly Journal of the Royal Meteorological Society, 137(656), 553–597. https://doi.org/10.1002/qj.828

Hersbach, H., Bell, B., Berrisford, P., Hirahara, S., Horányi, A., Muñoz‑Sabater, J., Nicolas, J., Peubey, C., Radu, R., Schepers, D., Simmons, A., Soci, C., Abdalla, S., Abellan, X., Balsamo, G., Bechtold, P., Biavati, G., Bidlot, J., Bonavita, M., … Thépaut, J. (2020). The ERA5 global reanalysis. Quarterly Journal of the Royal Meteorological Society, 146(730), 1999–2049. https://doi.org/10.1002/qj.3803

Szczypta, C., Decharme, B., Carrer, D., Calvet, J.-C., Lafont, S., Somot, S., Faroux, S., & Martin, E. (2012). Impact of precipitation and land biophysical variables on the simulated discharge of European and Mediterranean rivers. Hydrology and Earth System Sciences, 16(9), 3351–3370. https://doi.org/10.5194/hess-16-3351-2012

Kubota, T., Aonashi, K., Ushio, T., Shige, S., Takayabu, Y. N., Kachi, M., & Oki, R. (2020). Global Satellite Mapping of Precipitation (GSMaP) products in the GPM era. Satellite Precipitation Measurement, 355–373.

Gosset, M., Dibi-Anoh, P. A., Schumann, G., Hostache, R., Paris, A., Zahiri, E.-P., Kacou, M., & Gal, L. (2023). Hydrometeorological Extreme Events in Africa: The Role of Satellite Observations for Monitoring Pluvial and Fluvial Flood Risk. Surveys in Geophysics, 44, 197–223.

Peterson, P., Heim, R., Kaiser, D., & Kunkel, K. (2015). Climate Hazards InfraRed Precipitation with Station data (CHIRPS). Climate Data Guide.

9. The explanation provided in lines 513-525 and Figure 19 is somewhat vague and incomplete. The authors attempt to explain that the accuracy of the assimilation depends entirely on the uncertainty of the observations. However, the authors should first clarify the concepts of increment/innovation and residuals. Additionally, in the case of dH assimilation, a 0.4m observation error is compared to the dH uncertainty of the models, whereas in discharge assimilation, 30% of Qalti is compared with the modeled discharge. For discharge assimilation, model uncertainty is likely to be larger than dH uncertainty. Therefore, it is crucial to examine how a 0.4m variation in water level could affect the variation in discharge. Furthermore, the authors note that outliers at the peak were disregarded in the EnKF due to high observation errors, but they should also discuss how the EnKF would perform if an outlier occurred during the low-flow season. Additionally, the filter divergence and inflation applied may influence such behavior in the EnKF. Therefore, it is recommended that the authors further investigate this issue.

   Response :

We will clarify the concepts of increment/innovation and residuals when introducing Figure 19.

Regarding the impact of a 0.4 m variation in water surface elevation (dH) on discharge estimates, we acknowledge the importance of understanding this relationship. While we will not be able to conduct a dedicated sensitivity analysis, we can discuss this effect qualitatively based on the stage-discharge relationship. In river sections where the rating curve is nearly linear, a 0.4 m change in WSE would translate into a proportional change in discharge. However, in more complex settings, such as floodplains or low-flow conditions, where rating curves are non-linear, the same WSE variation could result in highly variable discharge adjustments. We will clarify this in the revised manuscript to provide better context on how WSE uncertainty impacts assimilation outcomes.

On the question of outliers and inflation strategies in the EnKF, we acknowledge their role in shaping assimilation results. In our experiments, outliers at peak flows were disregarded due to their high observation errors, which prevented them from influencing updates significantly. However, the impact of outliers occurring during low-flow conditions is less straightforward, as the relative uncertainty of observations and model states differs between flow regimes. A thorough investigation of this would require a dedicated study assessing how the EnKF processes extreme values across different hydrological conditions.

Similarly, analyzing the effect of inflation settings on the assimilation process would require a separate in-depth evaluation. Given the scope of our current study, we propose addressing these aspects in future work and will highlight these research directions in the manuscript's perspectives section.

10. Overall, the authors should enhance the presentation quality of the manuscript. The results section could be better structured by grouping similar ideas together, such as discussing discrepancies in the rating curves in a cohesive manner. Additionally, all figures should be improved, including more detailed captions that are self-explanatory. It would also be beneficial to include statistical measures such as NSE and KGE in the time series of discharge and water level. Furthermore, the authors should improve the citations and references; for instance, references like "?Biancamaria et al.," and "Filippucci et al. (in preparation)" should be clarified and properly formatted.

Response :

We thank the reviewer for the suggestions to improve the presentation quality of the manuscript. We recognise that certain aspects of the results section, figures, and references are to be improved for more clarity. We plan to address each of these points comprehensively in the revised manuscript, along with the related moderate and minor comments and the corresponding feedback from Reviewer 2.

Regarding the figures, we will improve their overall quality (better formatting for all figures, especially time series plots). We will revise the captions for Figures 3-4 (will be merged),

11 and 22, as highlighted in the minor comments (4, 5, 17 and 25) to make them more detailed and self-explanatory.

- Figure 3-4 (CTRIP / MGB) : "Schematic representation of the two large-scale hydrological models used in this study. Panel (a) illustrates the ISBA-CTRIP model (Decharme et al., 2019), a global river routing model coupled to the ISBA land surface model, which simulates river discharge at continental scale. Panel (b) presents the MGB model (Collischonn et al., 2007; Pontes et al., 2017), a semi-distributed hydrological model incorporating two-way coupling between hydrology and hydrodynamics, enabling the explicit simulation of floodplain processes such as evapotranspiration from inundated areas."
- Figure 11 (Experimental Setup): "Overview of the experimental setup for the assimilation of ESA CCI discharge products into the CTRIP and MGB hydrological models. The models were forced with ERA5 reanalysis data (GSMaP/IMERG precipitation data for MGB in the Niger and Congo Basins respectively) and run in both Open-Loop mode and with data assimilation. The three assimilated observation types are: (1) water surface elevation anomalies (dH) derived from satellite altimetry, (2) altimetry-based discharge (Qalti), and (3) discharge estimated from multispectral imagery (Qmultispec). The EnKF framework (HyDAS for CTRIP and HYFAA for MGB) was used to update model states based on these observations."
- Figure 22 : "Panel (a) presents the Kling-Gupta Efficiency (KGE) performance results from CTRIP for each assimilation experiment (colored markers) at 11 stations in the Niger Basin, with Open-Loop (OL) performance shown as grey triangles for reference. Panel (b) displays the corresponding discharge climatology at Niamey, comparing observed discharge (black) with Open-Loop simulations from CTRIP (grey) and different assimilation experiments (colored), illustrating seasonal variations in modeled and observed discharge. Panel (c) shows the rating curve at Niamey from CTRIP Open-Loop (black), compared to Obs CCI (dH, Qalti) (red)."

Additionally, we will incorporate statistical measures (NSE and/or KGE, depending on relevance) in discharge time series plots, either directly within the plots, similar to Figure 10 but with improved formatting for Figures 12, 13 and 14, or within the captions for Figures 17, 21, and 23.

Concerning the references and citations, we thank the reviewer pointing out formatting issues and missing references. We will correct the placeholder "?Biancamaria et al." in line 253 and replace the incomplete reference "Filippucci et al. (in preparation)" in line 801 with the proper citation.

**Moderate Comments:**

1. L4: HyDAS and HYFAA were not introduced. It is hard to think that the reader outside the field of river DA would know these acronyms.

   Response :

Lines 175-176 are changed with the following : "CTRIP-HyDAS (Hydrological Data Assimilation System) is a data assimilation framework integrated into the CTRIP model. Developed since the early 2010s, it is specifically designed to assimilate observations from satellite altimetry missions."

Line 185 is changed with the following : "HYFAA (Hydrological Monitoring and Forecasting Framework for Assimilation Applications) is a Python-based scheduler designed for hydrological monitoring and forecasting (Figure 6)."

2. L24: The authors would ideally need to start the discussion about the river DA in a broader sense such as by referring to relevant studies (e.g., Clark et al., 2008; Feng et al., 2021; Michailovsky et al., 2013; Paiva et al., 2013; Revel et al., 2023; Wongchuig et al., 2019). Then authors can narrow down the explanation to CTRIP and MGB DA.

Response :

We have revised lines 24–33 of the introduction to provide a broader discussion of data assimilation (DA) and its context in hydrological modeling. The revised section is as follows:

"Data assimilation has become a crucial tool in hydrological modeling, improving simulation accuracy by integrating observational data to update model states and parameters. This approach is particularly effective in addressing uncertainties in model inputs and structural representations, especially in large-scale applications. Clark et al. (2008) demonstrated how DA could enhance streamflow predictions through the assimilation of in-situ discharge measurements, significantly refining model forecasts. Michailovsky et al. (2013) extended this concept by incorporating remotely sensed water surface elevation (WSE) data into hydrological models, resulting in improved discharge estimates in regions with limited in-situ data.

Following these foundational studies, Paiva et al. (2013) investigated the assimilation of both in-situ and satellite-derived discharge data into the MGB model for the Amazon Basin, showing marked improvements in river flow simulations and flood forecasting. Wongchuig et al. (2019) further investigated the integration of simulated Surface Water and Ocean Topography (SWOT) mission data, demonstrating the potential of high-resolution satellite observations in refining hydrodynamic models. More recently, Feng et al. (2021) and Revel et al. (2023) have advanced DA methodologies by incorporating transformed WSE data, leading to notable improvements in discharge predictions and model performance. Building upon these developments, Wongchuig et al. (2024) introduced the Multi-Observation Local Ensemble Kalman Filter (MoLEnKF), which simultaneously assimilates various satellite-derived hydrological variables, demonstrating significant improvements in large-scale hydrological predictions.

In the context of large-scale hydrological models like ISBA-CTRIP and MGB, the integration of data assimilation (DA) frameworks has proven essential for mitigating uncertainties related to input parameters and model simplifications. Historically, many studies have focused on the added value of water surface elevation (WSE) data

assimilation into these models. For instance, Pedinotti et al. (2014) and Oubanas et al. (2018) demonstrated the significant benefits of assimilating WSE data to improve hydrological simulations. More recently, there has been a growing interest in the potential of discharge assimilation. Paiva et al. (2013) successfully combined discharge data with precipitation from TRMM to enhance the MGB model's simulations of Amazon Basin river dynamics. Similarly, Wongchuig-Correa et al. (2020) showed how SWOT discharge data could improve model accuracy in the Solimões and Negro river basins. Additionally, Emery et al. (2018) used altimetry-derived discharge data to refine simulations of river storage and discharge in large-scale models like ISBA-CTRIP. These efforts reflect a growing interest in leveraging discharge data to further enhance hydrological modeling.

Recognizing the importance of long-term, high-resolution data, the European Space Agency (ESA) initiated the Climate Change Initiative (CCI) Discharge Project to address the lack of long-term, high-resolution river discharge data. …"

3. L88: The rationale for selecting these two basins could benefit from further explanation, such as considerations of data availability, a lack of previous studies in the region, or other relevant factors. The motivation behind this choice should be emphasized more clearly.

Response :

We appreciate the reviewer's suggestion to elaborate on the rationale behind selecting the Niger and Congo River basins for our study. Our choice was guided by several key considerations:

For both basins, we have access to hydrological observations, including satellite-based data and in-situ measurements, which are essential for validation. While the Niger Basin has a relatively good availability of in-situ discharge records, the Congo Basin has seen a significant decline in active hydrological gauges since the 1960s (Tourian M.J. et al., 2023). This makes it an interesting case for evaluating the potential of satellite-derived discharge products in regions where ground-based data are sparse.

The Niger and Congo basins also present distinct hydrological characteristics, including differences in rainfall patterns, climate, tributary networks, and specific hydrological processes that are complex to model. The Niger Basin, for instance, includes an inner delta where intensive evaporation significantly alters streamflow dynamics, while the Congo Basin contains large lakes and. This diversity allows us to assess data assimilation performance across different hydrological contexts, broadening the relevance of our findings.

The Niger and Congo basins have distinct hydrological characteristics, such as variations in rainfall patterns, climate, tributary networks, and specific hydrological processes that are challenging to simulate. For instance, the Niger Basin features an inner delta with intensive evaporation, while the Congo Basin contains significant lakes and an extensive tributary network, where each contributes significantly to the discharge in the outlet. This diversity

allows for a comprehensive assessment of our data assimilation methods across different hydrological contexts, broadening the applicability of our findings.

Geographically, these rivers are two of the largest in Africa and play a crucial role in regional water resources, ecosystems, and human livelihoods. Understanding and improving their hydrological representation has direct implications for water management and policy decisions. However, despite their significance, these basins have been less studied in terms of advanced data assimilation techniques compared to other major river systems worldwide. Our study aims to address this gap by applying and evaluating data assimilation in the context of improving long-term river discharge representation for climate studies.

Flooding is another important factor in our selection. Both basins experience severe flood events with high socio-economic impacts. MGB, which is used for flood forecasting, provides an opportunity to examine the added value of data assimilation in improving discharge simulations, particularly at key stations like Niamey in the Niger Basin, where floods have historically caused major disruptions.

Lastly, the availability of existing hydrological models for these basins provides a strong foundation for this study. MGB has already been implemented for both the Niger and Congo basins, allowing us to build on prior work. CTRIP, as a global model, has been studied in the Congo Basin in a previous study (Munier et al. 2022).

These elements will be added in Section 2.1 when presenting both basins. We hope the additional information addresses the reviewer's comment.

References :

Tourian, M.J., Papa, F., Elmi, O. et al. Current availability and distribution of Congo Basin's freshwater resources. Commun Earth Environ 4, 174 (2023). https://doi.org/10.1038/s43247-023-00836-z

Munier, S., Boone, A., Biancamaria, S., & Le Moigne, P. (2022). Assimilation of SWOT discharge versus water level into CTRIP-12D over the Congo Basin. IAHS-AISH Scientific Assembly 2022, Montpellier, France, 29 May–3 June 2022. https://doi.org/10.5194/iahs2022-657

4. L216-222: In the paragraph, the authors presented the calculation of the "length of the localization" but how about the observation localization weight, w?

     Response :

We will refer to the work of Revel et al. 2019, as we used the same method and equations for determining the observation localization weight, w. We will clarify this in the manuscript to ensure proper reference  while keeping the level of detail concise.

5. Figures 9 and 10: Figures 9 and 10 can be combined. As Figure 9 shows the KGE and its components, those can be combined into Figure 10 by showing the numeric value of each metric.

Response :

Both figures will be put in supplementary material, please refer to our response to Major Comment #3.

6.  Section 4.1.3 would be more appropriate in the discussion section. Therefore, it is recommended to move it under "4.3 Discussion."

    Response :

We consider that this section is key to setting up the context for the data assimilation (DA) results that follow. The conclusions from Section 4.1.3 are directly tied to understanding the baseline model performance and identifying issues like rating curve discrepancies and biases, which are later addressed in the DA analysis. Moving it to the discussion would break the logical flow of the manuscript, so we'd prefer to keep it in its current position while making its connection to the DA results clearer.

7.  L483: The authors need to examine more about the mismatch between the model's rating curve and satellite-based rating curves such as parameter error, model physics, etc.

    Response :

We agree that the mismatch between the model's rating curve and satellite-based rating curves is an important aspect to consider. In our case, the satellite-based rating curve is derived purely from observational data (WSE from altimetry and in situ discharge), whereas the CTRIP model relies on Manning's equation, with empirical relationships used to determine certain parameters. For example, channel width is estimated using a mix of satellite imagery and empirical equations, while slope is derived from high-resolution datasets like MERIT-DEM. Similarly, the MGB-IPH model employs geomorphologic equations to estimate river width and depth based on drainage area, and applies the Manning equation for flow routing, with river slope obtained from digital elevation models like SRTM.

Given these differing methodologies, discrepancies between the model's rating curves and satellite-based rating curves are expected. A comprehensive analysis of these mismatches, including potential parameter errors and model physics considerations, would require an extensive investigation that is beyond the scope of the current study. . However, we acknowledge the relevance of this point and will mention in the discussion that future work could explore the specific impact of parameterization choices and model physics on rating curve discrepancies.

8.  L488-495: The authors should further investigate the spatial density, temporal frequency, and accuracy of the observations. This is crucial for future studies, real-time applications, etc. The authors could also provide recommendations on data

accuracy, spatial density, and other factors to improve the assimilation of discharge or water surface elevation (WSE).

Response :

We appreciate the reviewer's suggestion to further investigate the role of spatial density, temporal frequency, and observation accuracy in assimilation performance. These are indeed key factors that influence the effectiveness of data assimilation, and we fully acknowledge their importance. However, given the already extensive scope of this study, a detailed exploration of these aspects would go beyond the limits of the current paper.

That being said, we will highlight in the discussion and perspectives that future work should assess whether temporal frequency can compensate for lower spatial density in certain conditions, particularly by analyzing local data gaps at different stations. This would be relevant given the hydrological differences between the Niger and Congo basins, including the presence of an inner delta in the Niger Basin, a major lake in the upper course of the Congo Basin, and the unique structure of the Congo River's extensive tributary network, where each contributes significant water volumes.

Similarly, future studies could examine whether the required temporal frequency differs between WSE and discharge assimilation, particularly by assessing how quickly the model state reverts to its prior condition after assimilation. Another important avenue of research would be to quantify the impact of observation uncertainty on long-term bias correction, as well as its influence during high- and low-flow conditions, which we have already touched upon in our response to Major Comment 7.

We also want to clarify that our focus is not on real-time applications but rather on assessing how these datasets improve long-term hydrological modeling, with hydrological reanalysis in the context of climate studies being a longer-term goal.

We'll make sure to highlight these points in the revised manuscript, acknowledging their importance while keeping the study centered on its main analyses.

9. Figure 18: It would be better to compare the in-situ rating curves here, as the MGB model indicates that the locations shown in the figure exhibit hysteresis in the rating curve. The Qalti/dH does not seem to capture this hysteresis. Additionally, the authors should discuss the reasons for the differences in the rating curves between Obs_CCI and MGB.

Response :

To clarify the intent behind this figure and its focus within the study, the goal of Figure 18 is to evaluate the differences between the model's rating curves (MGB) with the satellite-derived CCI products being assimilated; this comparison is relevant as the differences noted directly affect the data assimilation outcomes. In-situ data, while used for statistical performance evaluation, are not relevant here since they are not the observations being assimilated, and in-situ WSE is unavailable to construct rating curves.

Our primary focus in Figure 18 is not on capturing hysteresis but rather on assessing the alignment of the general slope of the rating curves between the MGB model and the CCI observations. This slope reflects the broader relationship between WSE and discharge and has a direct impact on the assimilation process. Misalignment in the slope indicates fundamental differences in the representation of the river's state and parameters (roughness, width, slope), and shows the main challenges when assimilating only WSE into models with simplified river representations. Therefore, we believe that the absence of hysteresis in the CCI products is not the primary issue for WSE assimilation, as the general alignment and representation of the rating curves have a far greater influence on how WSE observations translate into discharge corrections.

10. L542-543: The authors can use the CCI discharge comparison with the in-situ observations (Section 2.4.2) to further reinforce the hypothesis of "The uncertainty for WSE ...". But that hypothesis contradicts the explanation given in the L551-554. So, the authors need to clearly explain the conditions and circumstances when discharge assimilation works well compared to dH assimilation. It will be helpful to investigate more about model parameters as well as the model physics when discussing the "preferred assimilation variable".

Response :

We appreciate the reviewer's suggestion to clarify the conditions under which discharge assimilation is more effective than WSE assimilation and to ensure consistency between our statements in lines 542-543 and 551-554. We recognize that our explanation may not have been clear enough in distinguishing when WSE uncertainty becomes a limiting factor and when discharge assimilation provides better results.

The key factor influencing whether WSE or discharge performs better in assimilation is not only observation uncertainty but also how well is the model's open-loop simulation performance. In CTRIP, where the open-loop performance is relatively poor downstream the inner delta (evaporation not being represented), WSE assimilation tends to perform better because the model gives more weight to observations with lower uncertainty. Since WSE data generally have lower uncertainty than discharge (0.2–0.4 meters for WSE versus 30–50% for Qalti), WSE observations are more strongly integrated into the analysis, leading to fewer residual errors after assimilation (as illustrated in Figure 19).

In contrast, in MGB, where the open-loop model is already more accurate thanks to evaporation representation and calibration, we see that even at stations where rating curves align well between model and observations (such as Malanville in Figure 17), discharge assimilation still provides better performance than WSE. This trend is also evident in the performance indices in Figures 15 and 16, which show that discharge assimilation consistently leads to greater improvements across most stations. This suggests that, when the model is not overly erroneous due to missing hydrological process representations, assimilating discharge directly improves streamflow simulations more effectively than WSE assimilation, which still requires conversion through a rating curve.

The comparison between CCI discharge and in-situ observations in Section 2.4.2 helps reinforce this interpretation. While CCI discharge products (Qalti and Qmultispec) capture seasonal variability well, they show varying degrees of bias and correlation across different stations. In CTRIP, where the model has significant biases, WSE assimilation is beneficial because it introduces fewer additional uncertainties. In MGB, even in cases where rating curves are well aligned, discharge assimilation still yields better results, confirming that direct Q assimilation tends to be more effective when the model structure is already well-calibrated.

The apparent contradiction between our statements in lines 542-543 and 551-554 comes from the need to separate two key effects: in CTRIP, WSE assimilation performs better due to the model's poor open-loop state downstream the inner delta, which allows lower-uncertainty WSE data to be assimilated more effectively; in MGB, even when rating curves align well, discharge assimilation remains preferable, as seen in the performance indices across stations.

We will revise the manuscript to clarify these points and explicitly discuss how model structure, open-loop performance, and observation uncertainty influence whether WSE or discharge is the preferred assimilation variable. We will also strengthen our discussion by better integrating the CCI discharge vs. in-situ comparison to reinforce our conclusions.

11. The high redundancy of the phrase "higher temporal resolution of Qmultispec" is evident throughout Section 4.3.2. The authors should summarize the content more concisely and present the key points clearly.

    Response :

We recognize that Section 4.2.3 contains redundancy, particularly with the repeated use of the phrase "higher temporal resolution of Qmultispec". We will revise the section to summarize the content more concisely and present the key points more clearly.

**Minor Comments:**

1. L20: Please provide the references to CTRIP and MGB

We will add references for CTRIP (Decharme et al., 2019) and MGB (Collischonn et al., 2007; Pontes et al., 2017) in the introduction.

2. L43-52: Why authors did not introduce Wongchuig et al., (2024)?

Please refer to response to Moderate comment #2.

3. Figure 1: It was felt that Figure 1, Figure 2, and Figure 8 can be summarized into one figure with two panels.

We plan to merge Figures 1 and 2. However, we prefer to keep Figure 8 near Table 1 for clarity. When revising the manuscript, we will evaluate whether merging all three figures preserves clarity, and if so, we will proceed with the suggested modification.

4. Figure 3 & 4: The authors need to include some more description on the captions of those figure.

Please refer to response to Major comment #10 .

5. Figure 4: In the caption of the figure "((Fleischmann et al., 2018))" should be "(Fleischmann et al., 2018)"

We will correct this formatting issue.

6. L173: The authors could combine Sections 2.2 and 2.3 considering CTRIP-HyDAS and MGB-HYFAA.

We prefer keeping these two sub-sections separate to ensure a clear distinction between the hydrological models before introducing their respective data assimilation schemes. This structure also benefits readers already familiar with CTRIP and MGB, allowing them to directly refer to Section 2.3 for details on data assimilation without revisiting model descriptions.

7. L219: the title of Section 2.3.2 can be revised as a broader title such as "Data assimilation method" rather than EnKF because the actual paragraph is presenting about LETKF.

We will modify the section title accordingly.

8. L216: "The localization" should be "The observations localization weight". In DA literature "localization" has a broader meaning than "w".

We will revise this terminology for clarity.

9. L217: "length of the localization" is commonly referred to as the local patch in DA studies which is the area used for collecting observations for assimilation for a given river pixel.

We will revise the terminology to align with DA literature.

10. L220: What is the threshold used for defining the "length of the localization"?

In our study, we determine the localization length for each river pixel by fitting a Gaussian semi-variogram over the variable. The range parameter from this fit indicates

the distance over which the variables (WSE anomaly or discharge) are spatially correlated. This range serves as the threshold for defining the localization length to make data assimilation accounts for spatial dependencies in the river system. This method is detailed in Revel et al. (2019). We will add these details in L220.

11. Figure 7: What does the color bar mean? observations localization weight (w)?

Yes, the color bar represents the observation localization weight. We will explicitly mention this in the figure caption.

12. L232: What is the meteorological forcing perturbated in making the ensemble?

Precipitation is the meteorological forcing that is being perturbed in making the ensemble.

13. Figure 9: The authors better reorganize this figure more carefully by adding figure panel numbers and titles like "Niger" and "Congo".

We will improve the figure layout by adding panel numbers and basin labels.

14. L253: Typo "?Biancamaria et al.,"?

We will correct this typo. Please refer to response to Major comment #10 .

15. L273: Check whether this method citation is correct "Filippucci et al. (in preparation)".

For Filippucci et al. (in preparation), if a published version or preprint with a DOI becomes available before the final submission, we will update the reference accordingly. Otherwise, we will remove it from the final version.

16. L275: Have any outlier removal methods been applied to the assimilated satellite data? Based on Figure 10, applying an outlier removal method could improve the accuracy of data assimilation.

Outlier detection and removal are planned for implementation in Phase 2 of the CCI Discharge project, as mentioned in line 660 within the perspectives section.

17. Figure 11: Some abbreviations, such as "Ppt" and "Pref," are not clear or defined in the caption. These should be clarified or spelled out for better understanding.

"Ppt" refers to precipitation, and "Perf." stands for performance. We will spell them out and define them clearly in the figure caption.

18. L335: The authors should introduce how anomalies were calculated in "water surface elevation anomaly (dH)".

Water surface elevation anomalies are computed by subtracting the mean water surface elevation from a 20-year reference (open-loop) simulation [2000-2020] in both models. We will clarify this in line 335.

19. Figure 12: It is better to show the KGE and NSE values in each figure.

We will add these values to the figure. Please refer to response to Major comment #10 .

20. Figure 13: Why does this figure not include CTRIP_OL

CTRIP_OL is already presented for the same stations in Figure 12 alongside MGB and observations. Since CTRIP performs poorly downstream of the inner delta, we removed it in all 4 panels of Figure 13 to allow for better visualization of MGB and observations. Please refer to lines 389-401 for further explanation.

21. Figures 15 and 16: These figures need further improvements, such as making the vertical axes comparable between the (1) and (2) columns and increasing the size of the horizontal axis labels for better readability.

We can only ensure comparable axes for MGB in panel (b). For panels (a), (c), and (d), maintaining the same axis scale is not feasible due to significant differences in performance scores (for instance, NSE is positive for MGB but negative for CTRIP in panels (a)). We will also increase the size of the axis labels to improve readability.

22. L501-503: The authors should discuss relative improvement/degradation from the open-loop.

We will expand this paragraph by including the percentage of relative improvement or degradation compared to the open-loop.

23. L533: Please revise the sentence structure to emphasize the idea.

We will revise L533 as follows : "This mismatch leads to EnKF divergence, resulting in outliers and highly negative NSE values, which makes dH assimilation unreliable at stations such as Kinshasa and Ouesso (Figure 21)."

24. Figure 17: Please confirm whether "Obs_CCI" refers to both WSE from satellite altimetry and altimetry-based discharge.

We confirm that "Obs_CCI" represents water surface elevation (WSE) from satellite altimetry in panels (1a,1c) and altimetry-derived discharge in panels (2b,2d). We will revise the caption to make this distinction clearer.

25. Figure 22: The authors need to explain more about the figure in the caption.

Please refer to response to Major comment #10

26. L557: What are the conditions to satisfy this such as "when the discharge estimations were accurate enough".

We will consider emphasizing the importance of discharge data accuracy by linking it to our previous comparison between CCI products and in-situ observations, as discussed in Section 2.4.2, as well as to the performance of the open-loop simulations. Please refer to suggestions made in response to Moderate Comment #10.

27. Figure 21: Typo "CCIP"

We will correct this typo : "Figure 21. Comparison of dH (1) and Qalti (2) Assimilation in Ouesso, Bangui, and Kinshasa: OL Simulation in black, Assimilation Results in blue and Observed **CCI** Data in red."

28. References: The references need to be corrected for example Biancamaria, S., et al. (2024). Water Surface Elevation Time Series Derived from Multiple Nadir Altimetry Missions: A Long-Term Dataset for Hydrological Applications. Hydrology and Earth System Sciences, In press.; Cassé, C., et al. (2015). Modeling of the Congo River basin hydrology: The CRBH model – Version 1.0. *Geoscientific Model Development, 8*(8), 2315–2333. https://doi.org/10.5194/gmd-8-2315-2015

We will carefully review and correct all references according to the correct citation format.

References :

Collischonn et al., 2007: Collischonn, W., Allasia, D., Silva, B. C., & Tucci, C. E. M. (2007). The MGB-IPH model for large-scale rainfall–runoff modelling. Hydrological Sciences Journal, 52(5), 878–895.

Pontes et al., 2017: Pontes, B., Monzo, P., Gauthier, N. C., & Salbreux, G. (2017). Membrane tension controls adhesion positioning at the leading edge of cells. Journal of Cell Biology, 216(9), 2959–2977.

---

## Author Comment (AC2)

**Responses to Comments of Reviewer #2**

**1. The current layout and structure of the paper need significant improvement.**

(1) The introduction of data and model should be divided into two separate sections.

(2) It's strange to include subsection 2.4.2 in the introduction of data and model.

(3) It's strange to include the discussion (i.e. subsection 4.3) in the section of Results.

(4) The layout should be improved, for instance, there are a large number of blank spaces between pages.

Response :

(1) Since we are moving Section 2.4.2 to the supplementary material, the data section will be reduced. Given the size of the remaining subsections, we prefer to keep the introduction of models, DA schemes, and observations together in Section 2.

(2) To improve readability and maintain focus on the core findings, and in line with Reviewer 1's suggestion, we are relocating Section 2.4.2 to the supplementary materials. This ensures that detailed evaluations of satellite-based discharges and the open-loop performance of the models remain accessible without diverting attention from the main narrative.

(3) We agree and will separate the Results and Discussion sections accordingly.

(4) We will also improve the layout to address the issue of excessive blank spaces between pages.

**2. The quality of presented figures need significant improvement.**

(1) There are too many figures presented in the paper, which I think the authors should try to reduce the number of figures to highlight the main results. For instance, I don't find the necessary to include the Figures 3-6, and Figures 1-2 and 8 can be merged into one figure.

(2) The quality of most figures should be improved, because it's difficult to read the text, numerical values and legend presented in most figures.

Response :

We appreciate the reviewer's suggestions to make the figures clearer and more concise.

(1) Figures 3 and 4 will be combined into a single figure, as we believe it's useful to keep a schematic of both models to highlight their key differences. For Figures 5 and 6, we will keep only Figure 5 to streamline the presentation.

(2) We understand the concern about readability and will improve the quality of all figures to ensure that text, numerical values, and legends are clearer and easier to read.

**3. It's not clear why the authors use both CTRIP and MGB models.**

Given the fact that the MGB model was already calibrated against in-situ discharge time series, I don't think it's fair to compare its performance to the CTRIP that was not calibrated yet. In addition, the input of precipitation for the two models are also different.

Response :

We thank the reviewer for their comment, which aligns with previous feedback from Reviewer #1 regarding the comparison of CTRIP and MGB. The purpose of this study is not to directly compare the absolute performance of these two models but rather to evaluate how the assimilation of long-term CCI discharge observations impacts their performance.

Both CTRIP and MGB are pre-existing modeling frameworks, each with its own set of parameters (calibrated or not), forcing datasets, and specific objectives. CTRIP is primarily designed for long-term climate studies and large-scale hydrological projections, whereas MGB is developed for operational forecasting at the basin scale. These differences in design naturally result in distinct open-loop performances. However, this diversity is precisely why we chose both models—to assess how discharge assimilation interacts with models that have different parameterization strategies, data inputs, and intended applications.

We acknowledge that MGB has been calibrated against in-situ discharge data, while CTRIP has not undergone formal calibration. However, CTRIP follows a physically based parameterization approach commonly used for global-scale hydrology, which does not necessarily rely on calibration. Similarly, the use of different precipitation datasets reflects the operational requirements of each model: ERA5 for CTRIP, which ensures global consistency for climate applications, and GSMaP/CHIRPS for MGB, which provides finer-resolution rainfall estimates suited for short-term forecasting.

In the revised manuscript, we will clarify these distinctions to ensure that our objective is not a direct model-to-model performance comparison but rather an assessment of how

long-term discharge assimilation affects two fundamentally different modeling chains. We will also explicitly state this limitation to enhance transparency.

**4. The design of DA experiments can be further enhanced.**

(1) The authors can consider implementing only the MGB model for the DA experiment, and I think the authors can consider two cases, the MGB model with and without calibration, to investigate the impact of calibration on assimilating satellite-based products.

(2) The authors can also consider assimilating the in-situ discharge data from the stations where both in-situ and satellite-based data are available, and the remaining stations can be used for the validation. As such, the subsection 2.4.2 can be included in the section of Results, and the impact of uncertainty of the satellite-based data can be further investigated compared to the performance of in-situ observations.

Response :

We appreciate the reviewer's suggestions on enhancing the design of the DA experiments. Indeed, a wide range of additional experiments could be conducted to further explore these aspects, but doing so would significantly extend the scope of the study beyond what is feasible for a single paper.

To ensure clarity, we are revising the introduction to better define the objectives and scope of this work. Our focus is on evaluating the impact of assimilating long-term satellite-derived discharge observations within two pre-existing large-scale hydrological models, each with its own structure and setup. While investigating the effect of model calibration on DA performance or incorporating in-situ discharge assimilation would be interesting, these aspects would require dedicated studies.

We hope this clarification helps situate our study within its intended scope.

**5. The Introduction can be further improved to review current progress of assimilating discharge data (including those research using in-situ observations) and relevant DA methods.**

Response :

We thank the reviewer for their comment on improving the introduction to provide a broader review of data assimilation (DA) progress, including studies using in-situ discharge observations and relevant DA methods. This suggestion aligns with Reviewer #1's moderate comment #2. In response, we have revised lines 24–33 of the introduction to

provide a broader overview of data assimilation (DA) and its context in hydrological modeling. The revised section is as follows:

"Data assimilation has become a crucial tool in hydrological modeling, improving simulation accuracy by integrating observational data to update model states and parameters. This approach is particularly effective in addressing uncertainties in model inputs and structural representations, especially in large-scale applications. Clark et al. (2008) demonstrated how DA could enhance streamflow predictions through the assimilation of in-situ discharge measurements, significantly refining model forecasts. Michailovsky et al. (2013) extended this concept by incorporating remotely sensed water surface elevation (WSE) data into hydrological models, resulting in improved discharge estimates in regions with limited in-situ data.

Following these foundational studies, Paiva et al. (2013) investigated the assimilation of both in-situ and satellite-derived discharge data into the MGB model for the Amazon Basin, showing marked improvements in river flow simulations and flood forecasting. Wongchuig et al. (2019) further investigated the integration of simulated Surface Water and Ocean Topography (SWOT) mission data, demonstrating the potential of high-resolution satellite observations in refining hydrodynamic models. More recently, Feng et al. (2021) and Revel et al. (2023) have advanced DA methodologies by incorporating transformed WSE data, leading to notable improvements in discharge predictions and model performance. Building upon these developments, Wongchuig et al. (2024) introduced the Multi-Observation Local Ensemble Kalman Filter (MoLEnKF), which simultaneously assimilates various satellite-derived hydrological variables, demonstrating significant improvements in large-scale hydrological predictions.

In the context of large-scale hydrological models like ISBA-CTRIP and MGB, the integration of data assimilation (DA) frameworks has proven essential for mitigating uncertainties related to input parameters and model simplifications. Historically, many studies have focused on the added value of water surface elevation (WSE) data assimilation into these models. For instance, Pedinotti et al. (2014) and Oubanas et al. (2018) demonstrated the significant benefits of assimilating WSE data to improve hydrological simulations. More recently, there has been a growing interest in the potential of discharge assimilation. Paiva et al. (2013) successfully combined discharge data with precipitation from TRMM to enhance the MGB model's simulations of Amazon Basin river dynamics. Similarly, Wongchuig-Correa et al. (2020) showed how SWOT discharge data could improve model accuracy in the Solimões and Negro river basins. Additionally, Emery et al. (2018) used altimetry-derived discharge data to refine simulations of river storage and discharge in large-scale models like ISBA-CTRIP. These efforts reflect a growing interest in leveraging discharge data to further enhance hydrological modeling.

Recognizing the importance of long-term, high-resolution data, the European Space Agency (ESA) initiated the Climate Change Initiative (CCI) Discharge Project to address the lack of long-term, high-resolution river discharge data. …"

We hope this addresses the reviewer's request.